# Laboratory studies of fresh and aged biomass burning aerosol emitted from east African biomass fuels-PART 1-Optical properties

**Damon M. Smith,[1,2,#] Marc N. Fiddler[3], Rudra P. Pokhrel[1], Solomon Bililign[1*]**

1. Department of Physics, North Carolina Agricultural and Technical State University, Greensboro, NC, 27411 USA,
2. Applied Sciences and Technology Program, North Carolina A&T State University, Greensboro, NC 27411, USA,
3. Department of Chemistry, North Carolina Agricultural and Technical State University, Greensboro, NC, 27411, USA

\# Current address: Department of Chemistry and Physics, Western Carolina University, Cullowhee, NC 28723
* Correspondence: Bililignsol@gmail.com; Tel.: (+13362852328);

**Abstract**: An accurate measurement of the optical properties of aerosol is critical for quantifying the effect of aerosol on climate. Uncertainties persist and results of measurements vary significantly. Biomass burning (BB) aerosol have been extensively studied through both field and laboratory environments for North American fuels to understand the changes in optical and chemical properties as a function of aging. There is a need for a wider sampling of fuels from different regions of the world for laboratory studies. This work represents the first such study of the optical and chemical properties of wood fuel samples used commonly for domestic use in east Africa. In general, combustion temperature or modified combustion efficiency (MCE) plays a major role on the optical properties of the emitted aerosol. For fuels combusted with MCE of $0.974 \pm 0.015$ referred to as flaming-dominated combustion, the single scattering albedo (SSA) values were in the range between 0.287 to 0.439, while for fuels combusted with MCE of $0.878 \pm 0.008$ referred to as smoldering-dominated combustion the SSA values were in the range between 0.66 to 0.769. There is a clear but very small dependence of SSA on fuel type. A significant increase in the scattering and extinction cross-section (with no significant change in absorption cross-section) was observed, indicating the occurrence of chemistry, even during dark aging for smoldering-dominated combustion. This fact cannot be explained by the heterogeneous oxidation in particle phase and we hypothesize that secondary organic aerosol formation is potentially happening during dark aging. After 12 h of photochemical aging, BB aerosol become highly scattering with SSA values above 0.9, which can be attributed to oxidation in the chamber. Aging studies of aerosol from flaming-dominated combustion, were inconclusive due to the very low aerosol number concentration. We also attempted to simulate polluted urban environments by injecting volatile organic compounds (VOCs) and BB aerosol into the chamber, but no distinct difference was observed when compared to photochemical aging in the absence of VOCs.

## 1 Introduction

The role of biomass burning (BB) aerosol on air quality, human health, cloud formation, and climate remain poorly quantified. BB aerosol play an important role in the earth's radiation budget and in the hydrological cycle by

absorbing and scattering sunlight and by providing nuclei for cloud condensation (Crutzen and Andreae, 1991). There have been several estimates of the radiative forcing of BB aerosol ranging from $0.03 \pm 0.12$ W m$^2$ (Forster et al., 2007) to the most recent estimate of $-0.2$ W m$^{-2}$ (Boucher, 2013). The uncertainty associated with the radiative forcing, is in the range of $-0.07$ to $-0.6$ W m$^{-2}$ (IPCC, 2014). This high level of uncertainty is associated with uncertainty in measuring the optical properties of BB aerosol (Andreae and Merlet, 2001;Koch et al., 2009;IPCC, 2014). In most cases, the measurements of aerosol optical properties are either limited to a specific source region or confined to a limited wavelength range. Internally versus externally mixed particles can have very different optical properties (e.g.,(Jacobson, 2000;Stier et al., 2006;Schwarz et al., 2008)). The processing of fire emissions leading to the eventual formation of secondary organic aerosol (SOA) is complex, including dilution, partial evaporation of the primary organic aerosol (POA) into gaseous species, photochemical reactions of organic species, partitioning of semi-volatile primary emissions into the condensed phase upon cooling, and multiphase chemical conversion (including cloud processing) (Bruns et al., 2016).

Furthermore, it is often wrongly assumed that the only two aerosol types that contribute significantly to light absorption on a global scale are black carbon (BC) and mineral dust. Current climate models fail to recognize that organic aerosol (OA) is not purely scattering (Bond et al., 2011;Ma et al., 2012;Bahadur et al., 2012;Laskin et al., 2015). Rather, there is a growing amount of data indicating that a certain class of OA, known as brown carbon (BrC), can be commonly found on a global scale, particularly in urban environments, where it contributes significantly to the total aerosol absorption- specifically in the lower visible and ultraviolet wavelength range, where BC absorbs weakly (Chung et al., 2012;Kirchstetter et al., 2004;Yang et al., 2009;Laskin et al., 2015). Global simulations suggest that this strongly absorbing BrC contributes from $+0.12$ to $+0.25$ W m$^{-2}$ or up to 19% of the absorption by anthropogenic aerosol (Feng et al., 2013;Brown et al., 2018;Saleh et al., 2015;Saleh et al., 2014).

The environmental and health costs of pollutants emitted from open biomass burning and cookstoves are significant and have been associated with human health effects, including early deaths and low infant birth weight. There is a strong evidence for acute respiratory illnesses such as asthma, and chronic obstructive pulmonary disease (COPD) associated with open biomass burning (Naeher et al., 2007;Stefanidou et al., 2008;Holstius et al., 2012;Johnston et al., 2012;Johnston et al., 2011;Elliott et al., 2013;Henderson et al., 2011;Delfino et al., 2009;Rappold et al., 2011;Sutherland et al., 2005;Smith and Pillarisetti, 2017). Wildfire can have health impacts well beyond the perimeter of the fire even thousands of miles downwind (Spracklen et al., 2009).

This work is focused on biomass fuels from east Africa. It is estimated that nine out of ten, or 573 million people in sub-Saharan Africa, will remain without access to electricity by 2030 (Bank, 2019). The African continent is the largest source of BB emissions, with recent studies estimating African contributions to be ~55% of total global emissions of BB aerosol (Ichoku et al., 2008;Roberts et al., 2009;Roberts and Wooster, 2008;Lamarque et al., 2010;van der Werf et al., 2010;Schultz et al., 2008). African combustion emissions are expected to grow. For example, organic carbon (OC) emissions from Africa, are expected to make up 50% of the total global emissions in 2030 (Liousse et al., 2014). Africa currently has the fastest growing population in the world; projected to more than double between 2010 and 2050 (UN, 2011).

BB is a global phenomenon, and it was shown that the long-range transport of pollutants emitted from BB can affect air quality very far from the source (Edwards et al., 2006;Williams et al., 2012). Although the optical properties of BB aerosol emitted by biomass species native to North America have been extensively investigated (Hodzic et al., 2007;Yokelson et al., 2009;Liu et al., 2014;McMeeking et al., 2009;Levin et al., 2010;Mack, 2008;Mack et al., 2010), biomass fuels native to sub-Saharan Africa have only been studied during a few field campaigns (Eck et al., 2001;Liousse et al., 2010;Formenti et al., 2003). Due to the very limited available data, the models being used for air quality and climate change in Africa rely on global inventories, which are primarily collected from North America, Europe and Asia (Bond et al., 2004;Streets et al., 2004;Bond et al., 2007;Klimont et al., 2009;Lamarque et al., 2010;Klimont et al., 2013), and are not consistent with satellite observations over Africa (Liousse et al., 2010;Malavelle et al., 2011;Liousse et al., 2014).

To our knowledge, laboratory studies of the optical properties of BB aerosol from solid wood biomass fuels common for domestic use in east Africa have not been conducted. The only other African fuel studied were savannah grass from South Africa during FLAME-4 (Pokhrel et al., 2016) and savannah grass from Namibia and Brachystegia spiciformis from Zimbabwe during the Impact of Vegetation Fires on the Composition and Circulation of the Atmosphere (EFEU) project (Hungershoefer et al., 2008). With the exception of these two examples almost all reported laboratory studies have been focused predominantly on North American fuels (Hodshire et al., 2019). To improve air quality and climate change models for Africa, there is a need for laboratory studies to measure optical properties of BB aerosol from African fuel sources as the aerosol age and interact with polluted air that has the same chemical profile as African megacities and rural areas.

Smog chambers provide a controlled environment for a comprehensive study of aerosol optical properties, chemical and morphological evolution, and SOA formation. While fuel specific studies cannot be easily compared to wildfire field studies (Akagi et al., 2012), they can be used to compare emissions from domestic biomass use where the fuel type is known and is often not mixed. It is suggested that burn conditions influence emissions and aerosol mass (Yokelson et al., 2013;Liu et al., 2017) and may be a key difference between laboratory and field studies. To extend the results from this kind of study to more realistic conditions, we compare our results to previous parameterization schemes. In our work, we use a tube furnace for initiating the burn, where we have full control of temperature, airflow, and material combusted. Comparative laboratory studies of BB aerosol optical properties using fuels from Africa and higher latitudes under varying conditions and background pollutant abundances and photochemical aging will provide information on factors most critical for radiative impacts of BB aerosol.

In the first part of his study, we report the results from three biomass fuels from east Africa considered for a systematic fuel-specific study of optical properties of BB aerosol under different aging and burning conditions using an indoor smog chamber. Optical properties were measured for BB aerosol produced under smoldering-dominated and flaming-dominated combustions for each fuel type. For each combustion condition, we report the measured optical properties (i.e. scattering and extinction cross-sections, and single scattering albedo (SSA)) for fresh emissions, dark aged, photochemically aged and photochemically aged, with added VOC's to represent urban emissions from a representative African megacity.

## 2 Experimental methods


For this study, authentic hard wood fuels were obtained from east Africa and left under a hood to dry out for over a year. The fuel moisture content was 10%. These samples were weighed on a calibrated analytical balance so that they would approximately yield a total aerosol loading representative of a scenario (urban, wildfire, etc.).

### 2.1 BB aerosol generation

For laboratory samples, BB aerosol were generated by combusting wood samples in a tube furnace. This process has been described elsewhere in detail and is summarized here for clarity (Poudel et al., 2017;Smith et al., 2019). Samples with a mass of 0.5 g were typically used for experiments, which generally produces enough BB aerosol

for optical property measurements without overloading any of the instruments. The mass loading was estimated by determining the total aerosol volume, obtained by measuring the volume distribution with a scanning mobility particle sizer (SMPS) and assuming a density of 1 g cm$^{-3}$ for fresh aerosol. Samples as small as 0.1 g and as large as 5 g have been used before, with the maximum mass loading for the tube furnace near 10 g. Biomass samples were placed in a quartz combustion boat (AdValue Technology, FQ-BT-03), which was in turn placed at the center of the working tube

inside the furnace (Carbolite Gero, HST120300-120SN). The tube furnace was set at 500 $^{\circ}$C and 800 $^{\circ}$C for each fuel. Unlike the heating coils used to initiate burning (Sumlin et al., 2018) where the temperature is uneven and localized, the tube furnace provided a uniform temperature throughout the sample as the tube provides a uniform heated region of 300 mm which is approximately the size of the quartz boat.

Oxygen content can be varied between ambient conditions and the oxygen-starved conditions found within

forest fires by mixing air from a zero-air generator (Aadco Instruments, 747-30) with nitrogen. Flows from both gases are regulated by calibrated mass flow controllers (MFC, Sierra Instruments). For this work, only zero air was used at a flow rate of 10 sL min$^{-1}$. Details of combustion characteristics including, fuel particle size, residence time and adiabatic combustion temperature are provided in the supplementary document (ST-2).

Modified combustion efficiency (MCE) were calculated from CO and $CO_2$ measurements. These

measurements underwent external calibration with either a pure gas (for $CO_2$) or a certified standard (199.7 ppm for CO and 5028 ppm for $CO_2$, purchased from Airgas National Welders). Gas filter correlation analyzers from Thermo Scientific were used to measure CO and $CO_2$ (models 48C and 41C, respectively). The change in the CO and $CO_2$ concentration was determined by comparing average measurements before a burn, and after the burn, once the measurements were stabilized. Averages of increase in concentration were taken soon after the measurements were

stabilized, but before dilution could take place. Between 80s and 300s measurements at 10 Hz were averaged for the pre-burn state, and ~300s for the post-burn state. MCE was determined by the following equation:

$$MCE = \frac{\Delta[CO_2]}{\Delta[CO_2]+\Delta[CO]} \tag{1}$$

The MCE for the 500 °C burn was 0.878 ± 0.008, which is hereafter referred to as smoldering-dominated combustion. For 800 °C burn case the average MCE was 0.974 ± 0.015, which is hereafter referred to as flaming-dominated combustion. The 0.5 g fuel produced about 600 to 800 µg m$^{-3}$ of mass loading in the chamber during smoldering-dominated combustion.

**2.2. Indoor smog chamber and characteristics**

        The North Carolina Agricultural and Technical State University (NCAT) indoor smog chamber has a volume of 9.01 m$^3$ and is lined by Fluorinated ethylene propylene (FEP) Teflon. Two sides each have a bank of 32 ultraviolet (UV) lights (Sylvania, F30T8/350BL/ECO, 36"), for a total of 64 lamps. Emissions from combustion (gas and
particles) were transferred to the smog chamber via heated (200° C), ¼ inch stainless steel tubing, after which they undergo cooling and dilution in a natural fashion rather than a stepwise process. A mixing fan was used to produces a well-mixed volume within 10 to 20 minutes after combustion. In these experiments, the fan ran for 10 minutes while smoke was being introduced to the chamber, then for another 10 minutes after the furnace had been disconnected from the chamber. The chamber was constantly diluted by zero air (from generator). The flow rate was varied depending
on the sampling demands of instrumentation but was usually around 4 L min$^{-1}$ for a normal cavity ring down spectroscopy (CRDS) experiment.

        The smog chamber was constructed to sample several particulate and gas-phase species. Ozone (O$_3$) was measured with a Thermo-Environmental Instruments UV photometer (model 49), and NO$_x$ was measured with a Monitor Labs fluorescence analyzer (model 8840). The O$_3$ and NO$_x$ analyzer signals were digitized with a DAQ
(National Instruments, USB-6002) and the signal was displayed and stored via custom software (LabVIEW). The measurements of O$_3$ and NO$_x$ were done only during the chamber characterization experiments and were not measured in subsequent experiments reported in this work.

        Several parameters concerning our chamber itself have already been determined and reported (Smith et al., 2019). Chamber performance is affected by the intensity and spectral character of radiation, surface-to-volume ratio,
and nature and condition of the wall surface (Hennigan et al., 2011). For our chamber, wall-loss rates of NO, NO$_2$, O$_3$ and PM were determined. Total light intensity was determined in a separate experiment by measuring photolysis of NO$_2$ and knowing the spectral output of the UV lamps. The wall loss rates for NO, NO$_2$, and O$_3$ were found to be $(7.40 \pm 0.01) \times 10^{-4}$, $(3.47 \pm 0.01) \times 10^{-4}$, and $(5.90 \pm 0.08) \times 10^{-4}$ min$^{-1}$, respectively. The NO$_2$ photolysis rate constant was $0.165 \pm 0.005$ min$^{-1}$, which corresponds to a flux of $(7.72 \pm 0.25) \times 10^{17}$ photons nm cm$^{-2}$ s$^{-1}$ for 296.0
– 516.8 nm, and the particle deposition rate was $(9.46 \pm 0.18) \times 10^{-3}$ min$^{-1}$ for 100 nm mobility diameter BB particles from pine (Smith et al., 2019). Total aerosol surface area peaks approximately 20 minutes after combustion, while total aerosol volume peaks approximately 45 minutes after combustion. The aerosol appears to be well mixed within 20 minutes of combustion, with the size distribution resolving into a single lognormal distribution. However, this distribution continues to shift towards larger particle sizes, even after remaining in the smog chamber for over 24 h,
as shown in Figure S3 for the smoldering-dominated combustion of Acacia and Eucalyptus. The gas and particle loss rates and other properties for our chamber are comparable to similar indoor smog chambers previously reported e.g.

(Babar et al., 2016;Leskinen et al., 2015;Wang et al., 2014;Paulsen et al., 2005)). The temporal evolution of total particle mass, CO, and $CO_2$ concentration is shown in Figure S2 for smoldering-dominated Acacia and Eucalyptus. Chamber pressure and temperature did not vary much from room pressure and temperature during these experiments.

Even when the chamber was clearly pressurized, our sensor was not sensitive enough to show a change in pressure. Chamber temperature started at room temperature (around 20 °C or slightly above) and increased to a maximum of 30 °C after 5 hours of use when all the UV lights were turned on, with most of the increase happening within the first hour (Smith et al., 2019). The temperature increase during photochemical aging can impact partitioning of the semivolatile components. The gas phase partitioning coefficient is temperature dependent though its value is only

available for a limited number of semivolatile compounds. We did not measure the impact of temperature on partitioning of the semivolatile components, but it is possible that the observed change in optical properties during photochemical aging could in part be attributed to this effect.

Growth of aerosol particle inside the chamber was represented as a growth in the geometric mean diameter (GMD) of the size distribution as shown in Figure 1. Aerosol growth in the chamber was expected to be due to

coagulation, diffusional losses of particles, and condensation of the gases into existing particles. It is evident from Figure 1 that growth was larger during photochemical aging conditions compared to the dark aging, indicating aerosol growth was due to condensation and subsequent chemical transformations. However, for flaming-dominated combustion, growth in GMD was the same during both dark and photochemical aging conditions indicating that there was no condensational growth in those experiments.

The chamber cleaning procedure is described in the supplement material ST-1.

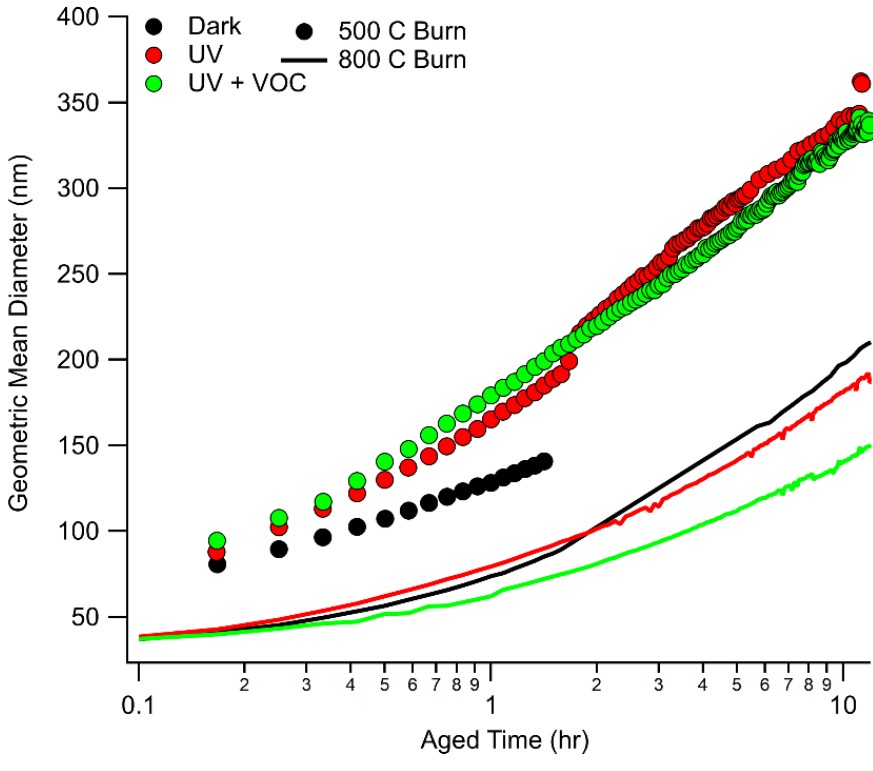

**Figure 1**: Growth of geometric mean diameter (GMD) as a function of photochemical age for eucalyptus under different burn and aged conditions represented in legend (black for dark aging, red for aging under light, and green for aging under light plus VOCs with solid line for flaming-dominated combustion and filled circle for smoldering-dominated combustion). Except for dark aging conditions, zero time represent the time at which light is turn on. For the lower temperature case, initial GMD was typically about 85-90 nm whereas for higher temperature burn it was below 40 nm.

### 2.3 BB aerosol aging

#### 2.3.1 Photochemical aging in a clean environment

For the purposes of these experiments, we define clean environment to be a smog chamber flushed out for 24 hours with clean air coming from the clean air generator. The only VOC's in the chamber came from the combustion of the fuel samples. Optical properties for fresh samples were measured using the procedure described below within 90 minutes of combustion, which allowed the size distribution of the aerosol to stabilize enough to conduct measurements without having a significant change in number density over the course of the experiment. For dark aging, the UV lights remained off, and measurements were repeated 12 hours after the introduction of BB aerosol in the chamber. For photochemical aging, a fresh sample was introduced into a clean chamber with the UV lights on immediately after combustion. Measurements were made after 12 hours of UV radiation.

#### 2.3.2 Photochemical aging in a polluted environment

To study the effect of photochemical aging in a polluted environment, a mixture of VOCs was used to simulate an urban atmosphere. These VOCs were injected into the chamber and allowed to mix with the chamber air while the furnace heated up before the introduction of fuel samples. The rest of this section describes the preparation of this VOC mixture.

A high degree of accuracy is required in setting up the conditions of an experiment and performing subsequent measurements. All gas-phase measurements were traceable to an analytical balance (calibrated yearly), NIST-certified flow meter (Mesa Laboratories, model Definer 220, calibrated yearly), NIST-certified stopwatch, and/or certified gas standard. Sample introduction accounted for the $NO_x$ produced from BB itself. Individual hydrocarbons and hydrocarbon mixtures were prepared with the analytical balance. These mixtures were composed of benzene ( ≥99.9%, Sigma-Aldrich), toluene (99.99%, Acros Organics), and ortho-xylene (99%, Alfa Aersar), and were prepared at the time of use. The concentration in molecules/cm$^3$ was determined consistently by measuring the mass of syringes before and after injection into the chamber. Using measured chamber pressure and temperature, the concentration in ppbv was estimated. All instruments were typically calibrated at the same time before a round of experimentation. NO and $NO_2$ were calibrated by passing certified standards through a calibrated MFC and mixing the standard with a calibrated flow of air in a ~30 mL glass mixing ball. Ozone was produced by passing air through an inline $O_3$ generator (UVP, model 97-0066-01). Using a calibrated $NO_x$ instrument, the $O_3$ mixing ratio was

determined by titrating it with NO to make $NO_2$. By measuring the $O_3$ signal, the calibration of $O_3$, in mV ppmv$^{-1}$, was performed.

        To represent a polluted urban environment, we used an emission inventory for urban environments from South Africa. This does not necessarily represent the east African emission inventory, but this does serve as a baseline, since it is the only available data to us for the continent. This data was obtained from the South African Air Quality

Information System (SAAQIS) and included concentrations of $NO_x$, NO, $NO_2$, CO, $O_3$, benzene, toluene, ortho-xylene, and ethylbenzene for several South African sites (Diepkloof, Kliprivier, Three Rivers, Sharpeville, Sebokeng, Zamdela, Thabazimbi, Lephalalae, Phalaborwa, and Mokopnae). The VOC data was obtained from the two weeks (M-F) of July 11 – 15 and July 18 – 22, which was in the middle of the peak burning season for South Africa for the year 2016. The urban areas (Diepkloof and Kliprivier) had combined average mixing ratios of 1.16, 3.48, and 1.44

ppbv for benzene, toluene, and o-xylene, respectively. Suburban areas (Three Rivers, Sebokeng, and Zamdela) had combined average mixing ratios of 1.69, 4.02, and 0.70 ppbv for the selected gases, respectively. Interestingly, suburban regions had somewhat higher average benzene and toluene mixing ratios, though o-xylene was only half the average urban concentration.

        A mixture was prepared using equal by volumes of benzene, toluene, and o-xylene, and 2.5 mg of the mixture

was injected by syringe into a U-shaped glass tube attached to the chamber. This resulted in a mixing ratio of 29.7, 24.9, and 21.9 ppbv for benzene, toluene, and o-xylene, respectively. The concentration injected into the chamber was approximately 7 – 26 times more concentrated than values found from urban South African emissions and 6 – 18 times more concentrated that suburban values. The reason for these elevated levels was mostly due to sample preparation constraints, since the amounts needed for an exact match were too small for our scale to weight appropriately.

Concentrations in the chamber were intentionally higher than atmospheric conditions, to age the BB aerosol faster and accentuate the potential effect of SOA.

**2.4 Optical properties measurement**

BB aerosol was size selected for optical property measurements by passing the sample through an impactor inlet with a 710 μm nozzle (3.8 μm diameter cut point), charge neutralizer (TSI model 3081), and a long differential mobility analyzer (DMA) (TSI model 3080). Particles with mobility diameters centered at 200, 300, and 400 nm were selected by the DMA for this study. We verified that the standard deviations of the size distributions did not overlap (Poudel et al., 2017). The aerosol number density was measured by a water condensation particle counter (WCPC)

(TSI model 3788), which was attached after the optical property instruments (shown in Figure 2) and provided flow through the entire setup at 0.58 sL min$^{-1}$. Further, the DMA and WCPC could be rearranged and combined to form an SMPS, which was used to determine size distributions before taking optical property measurements.

        Optical properties were measured using the extinction-minus-scattering technique (Weingartner et al., 2003;Bond et al., 1999;Sheridan et al., 2005), which uses CRDS to measure the total extinction of light and integrating

sphere nephelometry to measure the scattering of light for the same aerosol sample (Moosmüller et al., 2005;Thompson et al., 2002;Thompson et al., 2008;Strawa et al., 2006). The details of the CRDS/Nephelometry

optical properties measurement system is described in our recent work (Singh et al., 2014;Singh et al., 2016). A brief description is provided here. The extinction coefficient $\alpha_{ext}$ (m$^{-1}$) is > 1 and is defined as:

$$\alpha_{ext} = \frac{R_L}{c_{air}}\left(\frac{1}{\tau} - \frac{1}{\tau_0}\right) = \sigma_{ext}N_{CRD} \qquad (2)$$


where $c_{air}$ is the speed of light in air, $R_L$ is the ratio of mirror-to-mirror distance to the length of the cavity occupied by the sample, $\tau$ and $\tau_0$ are the ring-down times, or the time it takes the light intensity to reach 1/$e$ of the original intensity, of the sample and the blank measurement, respectively, $\sigma_{ext}$ is the extinction cross-section of the aerosol in m$^2$, and $N_{CRD}$ is the number concentration of the aerosol.

After being size selected, aerosol enter the ring-down cavity, where extinction was measured by passing a laser beam coupled to the cavity mode through the sample volume. The 355 nm beam from a Continuum Surelite I-20 Nd:YAG laser (at 20 Hz) pumps an optical parametric oscillator (OPO) laser, which can produce a range of wavelengths. For this study, we used a wavelength range of 500 to 570 nm and a single set of mirrors. Highly reflective mirrors confine most of the laser intensity within the CRDS, with a photomultiplier tube measuring the intensity of

the light exiting the mirror after each round trip inside the cavity. Extinction is determined from the decay of light intensity exiting the mirrors. Our system allows measurement of optical properties at a wide range of wavelengths over most of the solar spectrum to determine "featured" extinction cross sections as a function of wavelength.

A purge flow of nitrogen was used to keep the mirrors clean. After the CRDS the aerosol enter the integrating nephelometer (TSI, model 3563), where scattering is measured at three wavelengths (centered at 453, 554, and 698

nm). Lastly, the number density of the particles was measured by the WCPC, as stated above. As previously reported by Singh et al. (2014), estimated particle losses in the CRDS are 14.2, 14.7, and 11.4% for 200, 300, and 400 nm particle sizes, respectively, and estimated losses in the nephelometer are 8.6, 7.1, and 6.3 % for the aforementioned sizes, respectively.

All flows, except the DMA sheath flow, are calibrated against a NIST-certified flow meter (Mesa Laboratories, model Definer 220) that is factory calibrated on a yearly basis and has a listed accuracy of < 1 %. Figure 2 describes the flowchart of the experiments.

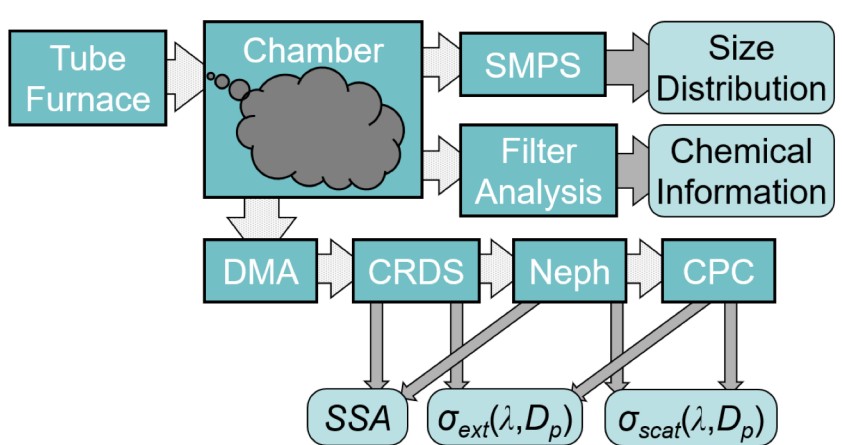

**Figure 2**: Scheme and flowchart for optical properties measurement


**2.5 Error analysis of optical properties measurements**

In our previous work (Singh et al., 2014) we have comprehensively and holistically accounted for known sources of random and systematic errors and developed a statistical framework for including the contributions to random error. The combined extinction cross section uncertainty (10 − 11%) was largely dominated by CPC measurement error (10%). The calculation flow for determining the average extinction cross section ($\sigma_{ext}$), absorption cross section ($\sigma_{abs}$), and single scattering albedo ($\omega$), was already described (Singh et al., 2014). The estimated

uncertainties are 11%, 15%, and 2.1% for $\sigma_{ext}$, $\sigma_{abs}$, and $\omega$, respectively.

The DMA can often allow multiply charged particles to pass through, that can result in artificially large measured cross sections, even for small number densities (Uin et al., 2011). Other groups have shown that measured extinction coefficients exceeded the predicted ones for 100 and 200 nm particles, which are most affected by the "multiple size–multiple charge" problem (Radney et al., 2009). As such, only particles 200 nm or greater were

considered in this work. However, even with 200 nm particles, it has been shown that a small DMA sizing error can still produce significant changes in the extinction (Radney et al., 2013). In principle, errors in the DMA must be corrected (Miles et al., 2011;Toole et al., 2013). However, we did not make corrections due to DMA sizing error in this work but reported the percentage errors (the overestimation of SSA) caused by multiply charged particles for each aerosol size studied.

**2.6 Aerosol chemical speciation monitor (ACSM)**

An aerosol chemical speciation monitor (ACSM; Aerodyne Research Inc., USA) was used to measure the chemical composition of sub-micron non-refractive particulate mass. Details about the ACSM can be found elsewhere (Ng et al., 2011). Briefly, dry aerosol from the chamber was sampled into the ACSM through a critical aperture with a diameter of 100 µm at a flow rate of 85 mL min$^{-1}$. The recorded data was processed using the ACSM local toolkit

(v.1.6.0.3) for Igor Pro. Since this work does not use mass loading in a quantitative way, we chose a collection efficiency of 1 for all species, similar to a previous study (Ng et al., 2011).

**3 Results and discussion**

**3.1 Optical properties measurements.**


Aerosol optical properties, namely scattering and extinction coefficient for size selected aerosol, were measured for three different east African biomass fuels. The selected fuels (eucalyptus, olive, and acacia) represent the most common trees in east Africa for domestic use, which contributes to significant aerosol loading. Each fuel was placed into the tube furnace at two different combustion temperatures. The term combustion temperature for this

work represents the set temperatures of the furnace (initially heated at 500 ℃ and 800 ℃) to investigate the impact of ignition temperature on aerosol optical properties and chemical composition. In both cases, the furnace reached the desired temperature before the sample was placed inside. In this paper, only the results from smoldering-dominated

combustion under all aging conditions and the results of the fresh flaming-dominated combustion samples are reported. SSA values were measured for size selected particles having mobility diameters of 200, 300 and 400 nm.


**3.2 Impact of size on SSA**

SSA was calculated by taking the ratio of scattering coefficient to the extinction coefficient measured for the wavelength range from 500 – 570 nm at 2.0 nm interval. Calibration of the system and the error analysis in the

calculation of SSA from the experimental measurements is described in section 2.5. We calculated the scattering coefficients at the CRDS wavelength range by using the scattering angstrom exponent from the measured scattering coefficients.

Submicron aerosol show size dependent SSA values in visible wavelengths. Size-selected SSA values measured at 532 nm in this study are compared with the size selected SSA valued calculated using Mie theory for two

different refractive indices as shown in Figure 3. These refractive indices were proposed by Bond and Bergstrom, (2006) for BC and Levin et al. (2010) for smoldering biomass burning particles. While no pronounced size dependence was observed for aerosol from flaming-dominated combustion, contrary to what was predicted by Mie theory, the SSA did show size dependence for aerosol from smoldering-dominated combustion. This is due to the impact of multiply charged aerosol not discriminated by the DMA. Since we did not correct for the presence of multiply charged

particles in this work, as described in section 2.5, this behavior was expected. The impact of multiply charged particles is more significant for flaming-dominated combustion (no pronounced size-dependent SSA) compared to the smoldering-dominated combustion (with some size-dependent SSA). This is a consequence of different particle size distributions. As shown in Figure S3, the presence of a second peak in the size distribution for flaming-dominated burns was expected to increase the impact of multiply charged particles on the observed SSA.

Recently, we estimated the impact of multiply charged particles on SSA using an Aerosol Particle Mass Analyzer (APM; Kanomax model 3602). We did not have this capability early in this study. APM was connected in-line after the DMA and subsequent optical properties were measured for freshly emitted aerosol. Details about the measurement strategy is given by Radney and Zangmeister (2016). It was found that, for smoldering-dominated burns, our SSA values were over estimated by a maximum of 8% for 200 and 300 nm sizes and by 5% for 400 nm size

particles. Whereas, for flaming-dominated burns, our SSA values were overestimated by 12% for 200 and 300 nm sizes and by 9% for 400 nm size.

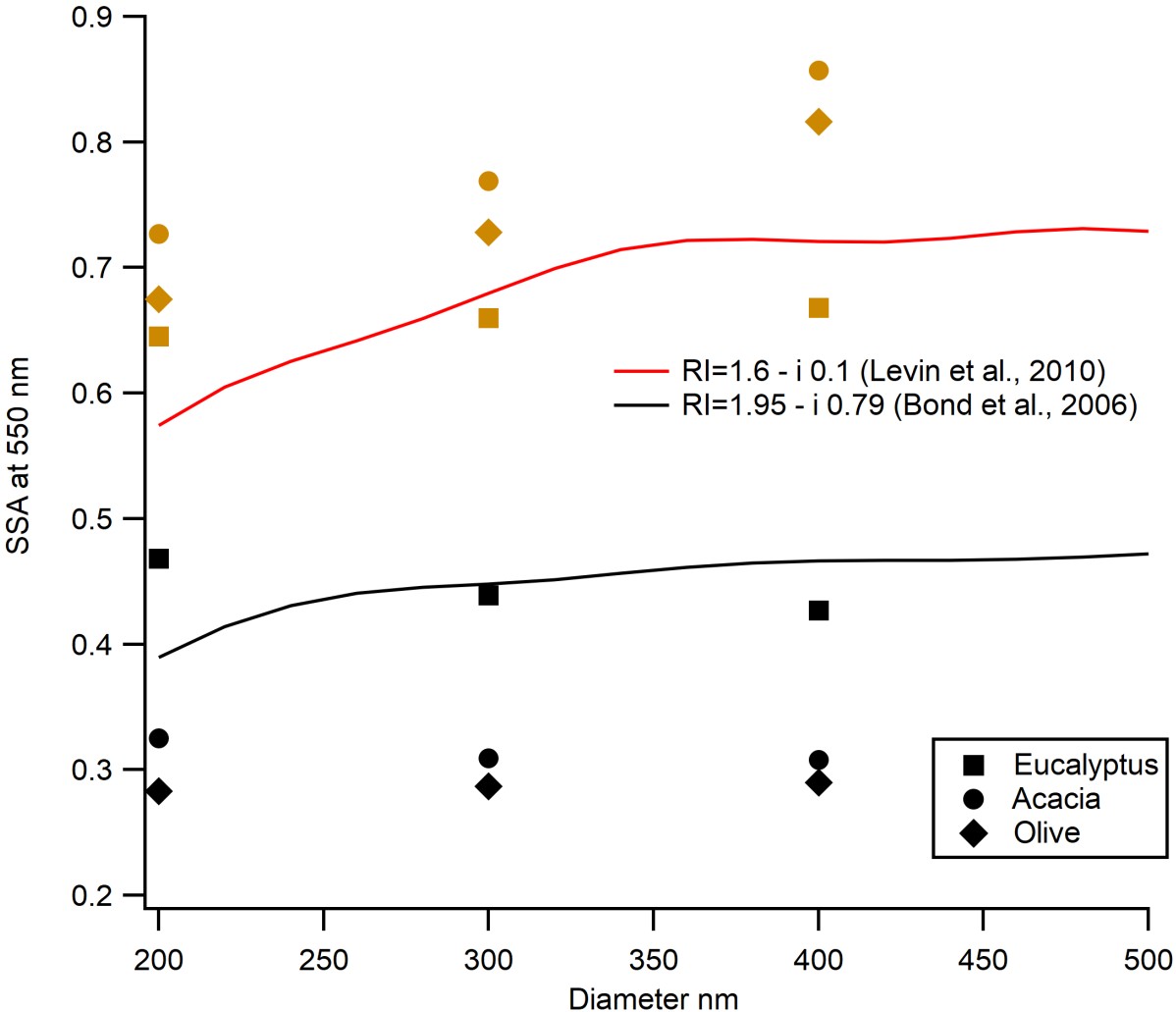

**Figure 3**: Impact of aerosol particle size on SSA. The solid red line is modeled SSA using Mie theory using refractive index from Levin et al. (2010), representing typical BB emission and black line is modeled SSA by Mie theory using refractive index from Bond et al. (2006), representing black carbon. Neither line is a fit to the data. Symbols are the different fuel types represented in the legend with black color for flaming-dominated combustion and brown color for smoldering-dominated combustion.

**3.3 Single scattering albedo of freshly emitted aerosol**

Figure 4 shows plot of SSA vs wavelength of light for freshly emitted 300 nm size aerosol from the different fuels at two different furnace temperatures (MCEs). The 200 and 400 nm particles show a similar behavior. The results show no wavelength dependence of SSA in the measured wavelength range of 500 −570 nm at both combustion temperatures. The dashed lines represent the propagated uncertainty (1 standard deviation) of the SSA, based on extinction and scattering coefficients. The extinction errors from the CRDS are mainly influenced by variability in the ring-down time. The SSA shows dependence on fuel type even under the same combustion condition. This was consistent for all particle sizes investigated.

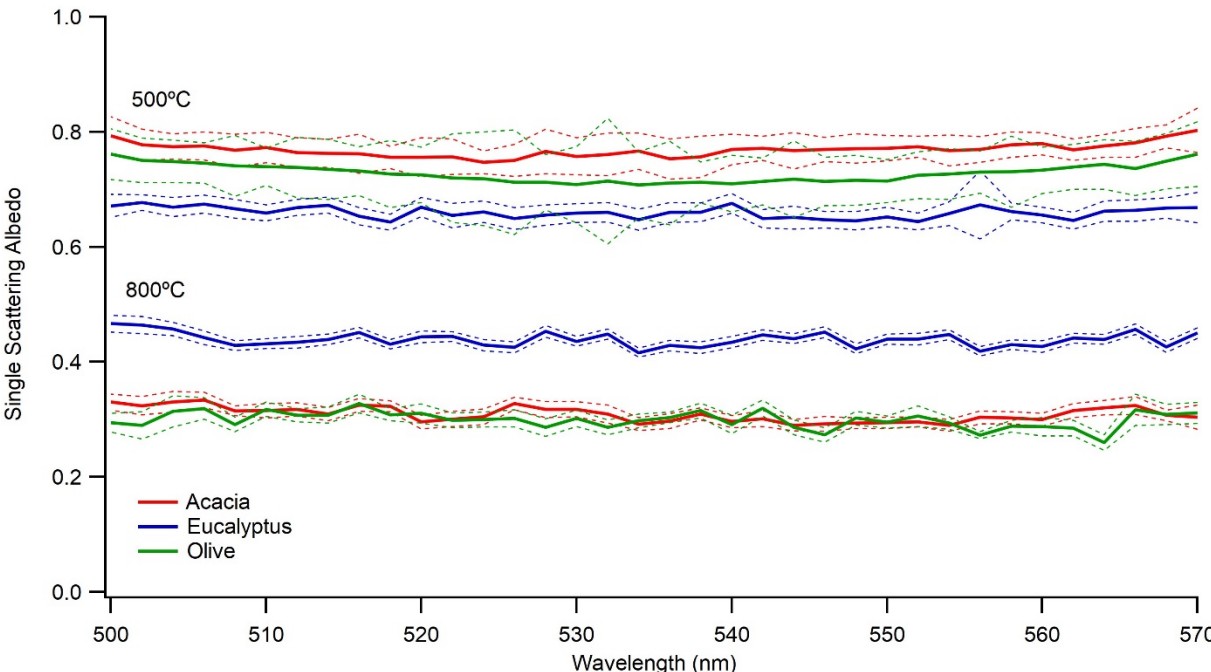

**Figure 4:** Single scattering albedo of 300 nm size-selected aerosol emitted at combustion temperatures of 500 °C and 800 °C. Solid blue, green, and red lines are for the average SSA from the three measurements and the dotted lines are the corresponding uncertainties (1 standard deviation) in the measured SSA.

The SSA values of the flaming-dominated combustion ranges from 0.287 to 0.439, whereas the SSA values of the smoldering-dominated combustion ranges from 0.66 to 0.769 for the different fuels. The average MCE for flaming-dominated combustion was 0.974 ± 0.015, while the average MCE for smoldering-dominated combustion was 0.878 ± 0.008. These MCE values suggest that at 800 °C combustion is flaming dominated which produces more BC, and at 500 °C combustion is smoldering dominated, which produces more OC (Christian et al., 2003;Ward et al., 1992). This explains the lower SSA values at higher combustion temperatures. A qualitative visual measure of the impacts of combustion temperature on aerosol properties can also be gleaned by looking at the color of the collected filter samples, as shown in Figure S4. As evident from Figure S4, flaming- dominated aerosol look black, whereas smoldering-dominated aerosol look brownish, indicating a visual difference between BC dominated and OC dominated emissions from the same fuel under different combustion temperatures.

The range of SSA for smoldering-dominated combustion was comparable to previous studies with similar MCE values (Liu et al., 2014;Pokhrel et al., 2016). On comparing the SSA of the three different fuels under two different combustion temperatures, it is apparent that SSA is controlled more by combustion condition than fuel type. Although there appeared to be a small but clear dependence of SSA on fuel type, there was larger variation in SSA for the same fuel under two different combustion conditions, compared to the variation due to the inter fuel variability under the same combustion temperature. This result was consistent with a previous study, which showed that SSA is highly correlated with the ratio of elemental carbon to total carbon (proxy for the combustion condition), even for a

wide variety of fuels (Pokhrel et al., 2016). A complete list of sizes selected SSA of fuels measured at two combustion temperatures and under different aging conditions is provided in Table S1.

In the companion paper to this (Smith et al., 2020) methanol extracts from BB aerosol collected on Teflon filters were analyzed by ultra-performance liquid chromatography interfaced to both a diode array detector and an electrospray ionization high-resolution quadrupole time-of-flight mass spectrometer (UPLC/DAD-ESI-HR-

QTOFMS) in negative ion mode. This was used to determine the relative abundance and light absorption properties of BB organic aerosol constituents. MS analysis of BB aerosol extracts from flaming-dominated combustion (Figure S5a and Table S2) revealed very little difference between the two fuel types, suggesting that there are either very few smoldering-dominated aerosol species produced for either fuel under these combustion conditions, or there are numerous species that are essentially the same between the samples. However, given that Eucalyptus has a higher

SSA than Acacia, this would suggest that Eucalyptus has more non-absorbing OA, or at least less absorbing than BC. Since it was Acacia that appears to have many more low-abundant organic constituents, several possibilities exist to explain these differences in SSA, as explored in more depth in the companion paper (Smith et al., 2020). It is likely that Eucalyptus combustion products were not captured by some aspect of the extraction and UPLC/DAD-ESI-HR-QTOFMS analyses, that the observed differences in SSA are due to morphology differences, or some combination

thereof. One potential explanation would be the presence of significant amounts of eucalyptol in the BB aerosol, which is a large fraction of Eucalyptus oil, and is a cyclic ester that lacks any basic functionality amiable for negative ion mode analysis, has good solubility in alcohols, and does not absorb in the UV and visible spectrum. An examination of the UV-visible spectra (Figure S6a) from the DAD shows no absorbing species in either region.

Chemical analysis revealed that for smoldering-dominated combustion, Eucalyptus and Acacia had a variety

of compounds in common, such as lignin pyrolysis products, distillation products, and cellulose breakdown products (Figure S5b and Table S2). Several lignin pyrolysis products and distillation products are more prevalent in Eucalyptus than in Acacia, while pyrolysis products of cellulose and at least one nitroaromatic species were more prevalent in Acacia. Given that these lignin pyrolysis and distillation products are known chromophores and are more prevalent in Eucalyptus than in Acacia, while Acacia has a higher abundance of non-chromophores derived from sugars and

cellulose, one would assume that Eucalyptus would be more absorbing (i.e. have a lower SSA) than Acacia in the visible spectrum. Despite the chemical analysis not capturing absolute amounts of OA, Acacia was found to have an SSA that is higher than Eucalyptus by 0.1 to 0.2, which is consistent with chemical measurements. This suggests that Acacia has either larger absolute amounts of non-chromophore compounds or Eucalyptus has a greater quantity of chromophores whose absorptive properties extend to the 500 – 570 nm region of the visible spectrum. An analysis of

the chromatographically-integrated UV/Visible spectrum (Figure S6b) shows that there are chromophores whose absorption features peak near ~290 nm and extend into the 500 – 570 nm region, though a normalized spectrum does not appear to show drastic differences between species.

Figure 5 shows the SSA plotted as a function of MCE at 532 nm. Overall, our values of SSA agree well with the previous studies (Pokhrel et al., 2016;Liu et al., 2014) with some outliers. This could potentially be because we

are comparing results for size-selected as opposed to bulk aerosol. In general, the variation of the SSA with MCE from the size-selected and bulk aerosol show a consistent behavior with higher SSA for the lower MCE cases and

lower SSA for higher MCE cases. As mentioned earlier, there occur some variabilities in SSA and MCE values even for the same combustion temperature. This could be due the dependence of SSA and MCE on fuel type or due to factors that we are not aware of. In general, however, combustion temperature plays a major role on the optical properties of the emitted aerosol. This suggests that by simply varying the combustion temperature, we can generate aerosol with very different optical properties and combustion efficiencies  (Saleh et al., 2018;Liu et al., 2014).

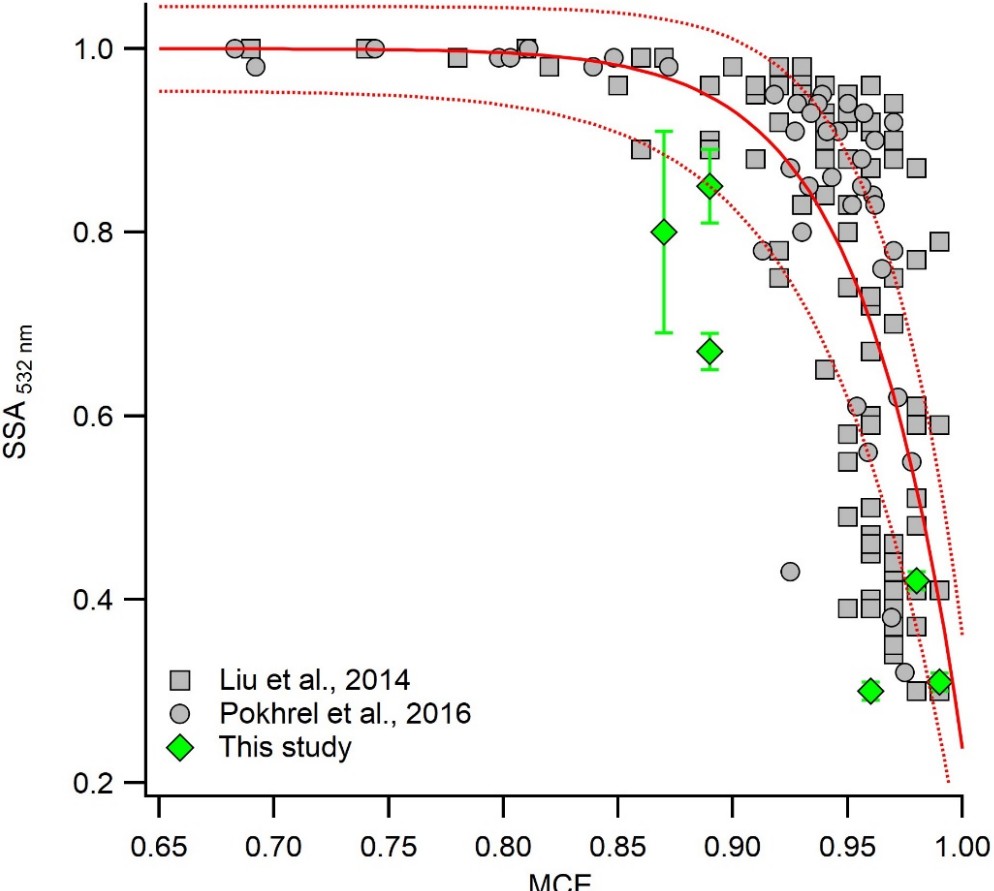

**Figure 5**:  SSA of 400 nm size-selected aerosol at 532 nm as function of MCE under different combustion temperatures. Gray symbols are the SSA of bulk aerosol from previous studies (Liu et al., 2014; Pokhrel et al., 2016). Solid and dashed red lines are the best fit and the uncertainty bounds proposed by  Liu et al. (2014).

### 3.4 Impact of dark aging on SSA

As BB aerosol age, their properties evolve due to competing chemical and physical processes (Hodshire et al., 2019;Yokelson et al., 2009;Akagi et al., 2012;Vakkari et al., 2018;Formenti et al., 2003;Garofalo et al., 2019). The particle dynamics was different in the dark compared to photochemical aging since there was, a pronounced increase of particle size and density which was also observed in previous laboratory and ambient measurements (Reid et al., 1998;Zhang et al., 2011). We did not have the capability to measure density during this work, however Figure S1 provides the number density as a function of time for smoldering-dominated Acacia and Eucalyptus as they aged

in the dark. Even though the RH remained the same and low in our experiments, under high relative humidity conditions, heterogeneous reactions may be facilitated to produce more water soluble inorganic salts such as sulfates and nitrates (Shi et al., 2014). The first nighttime field analysis of BB plume intercepts for agricultural fuels showed that oxidation for rice straw and ponderosa pine was dominated by $NO_3$ (Decker et al., 2019). To simulate the impact of dark aging on aerosol optical properties, BB aerosol was aged in dark for 24 hours in absence of UV lights and

additional ozone. The relative humidity in these experiments was very low (i.e. below the detection limit of our instrument). Optical properties of the freshly emitted aerosol were measured initially, with repeat measurements taken after the particles were left in the chamber to age in the dark. Figure 6 shows the impact of dark aging on SSA for the 300-nm size-selected aerosol emitted during smoldering-dominated combustion. Regardless of fuel type, there occurred an increase in SSA during aging with some fuel dependence, the largest of which was observed for olive. A

two-tail T-test confirmed that increase in mean SSA is statistically significant for all three cases. The results are similar for the 200 nm and 400 nm particles as shown in Table S1.

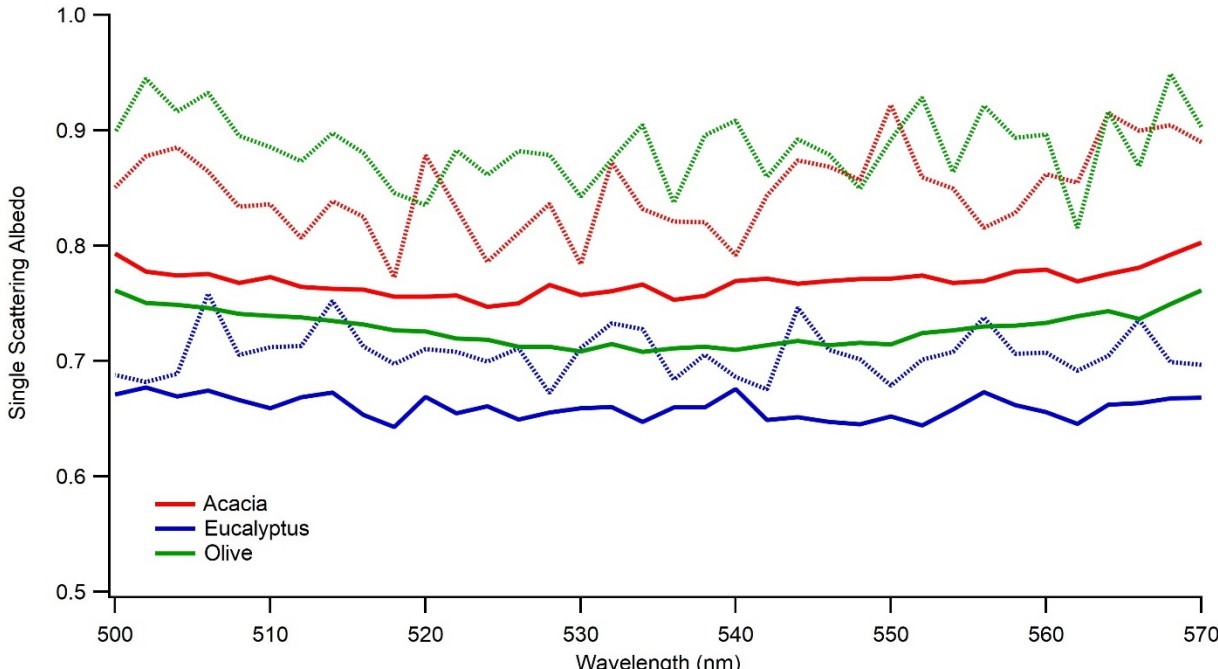

**Figure 6**: Impact of dark aging on SSA of 300 nm sized-selected aerosol emitted during smoldering-dominated combustion. Solid
lines are for freshly emitted particle and dotted lines are for dark aged particles in the chamber. Different colors are for the different fuels listed in the legend.

Nighttime chemistry in BB studies is still unclear (Hodshire et al., 2019). The potential mechanism for the observed result could be due to the formation of less/non-absorbing secondary organic aerosol. To further explore the

possibility of the observed increase in SSA, we looked at the scattering and extinction cross-section of the fresh and dark aged aerosol. Figure S7 shows the changes in extinction and scattering cross-section of 300 nm size particles emitted during smoldering-dominated combustion under dark aging. For all fuel types studied, there occurred a

significant increase in the scattering and extinction cross-section, indicating the occurrence of chemistry, even during dark aging. The increase in cross-section was driven by the scattering cross-section, with no significant change in absorption cross-section during aging. If a greater portion of the particle consisted of scattering SOA, one would expect a decrease in absorption cross-section if scattering cross-section of the particle increase for a given particle size, but this did not seem to be the case. While we did not characterize the chemical constituents of dark aged BBA in these experiments, there is a significant body of literature that has. (Li et al., 2015;Ramasamy et al., 2019;Hartikainen et al., 2018). Previous work by Tiitta et al. (2016) investigated dark and UV aging under several combustion conditions. It is not entirely clear which is equivalent to our smoldering-dominated combustion cases, given that their "slow ignition" produced significantly more VOCs when compared to their "fast ignition" experiments, but more hydrocarbon-like POA was produced from "fast ignition" experiments. These experiments were not distinguished by a combustion temperature and the MCE was not measured, but they were differentiated based on the amount of kindling used in their masonry heater. In either case, a significant amount of organonitrates were formed during dark aging via oxidation from $NO_3$ in the presence of ozone. Only in the slow ignition case was there a significant formation of SOA from ozonolysis during dark aging (~12% by mass from ozonolysis compared to ~12% from OH radical $NO_3$ and ~76% from and proxy radical based on positive matrix factorization analysis). A significant increase in the inorganic fraction of the aerosol was not expected given the low RH in these experiments (Li et al., 2015). Hartikainen et al. (2018)  also observed significant formation of nitrogen-containing organic compounds in both particulate and gas phase during dark aging, with significant partitioning into the condensed phase. Little oxidation of the particulate phase was observed compared to photochemical aging. While these studies examined chemical transformations in the particulate and gas phases, they did not characterize the effects of these changes on optical properties. Given the increase in the scattering cross-section of the particles without altering the absorption cross-section observed in this work, one likely explanation is that smaller particle with high fractions of BC increase their mobility diameter upon SOA formation. This SOA, however, is partially absorbing, so that the expected decrease in absorption was not seen; or was at least mitigated to a change within uncertainty. While this absorption must extend to the 500 – 570 nm region of the spectrum, several nitro-aromatic species and functionalized PAHs are able to absorb at such long wavelengths (Fleming et al., 2020).

Unlike smoldering-dominated combustion, we were not able to track the aging of aerosol emitted during the flaming-dominated combustion due to some experimental issues. First, there was a significantly low aerosol emission due to more complete combustion of the fuel. For 300 nm and 400 nm size ranges, the number concentration of the particles emitted during flaming-dominated combustion was a factor of two to four lower than those emitted during smoldering-dominated combustion. In addition, due to the highly absorbing nature of the flaming-dominated aerosol, the scattering cross-section of the aerosol was significantly lower than that of the smoldering-dominated aerosol. Therefore, due to the very low number concentration and highly absorbing nature of the particles, the scattering coefficient of flaming-dominated aerosol was below the detection limit of our nephelometer during the aging experiments. Hence, we did not feel confident in reporting the SSA of dark aged particles emitted during flaming-dominated combustion. Figure S8 shows the impact of dark aging on the extinction cross-section of flaming-dominated aerosol. Even though a significant increase occurred in the extinction cross-sections during dark aging of

smoldering-dominated aerosol, we did not observe such behaviors during aging of flaming-dominated aerosol, indicating no significant changes in their optical properties. As shown in Figure S8b, there was a slight increase in the extinction cross-section for olive. However, when accounting a 12 % uncertainty in the cross-section, this increase in extinction cross-section is statistically insignificant. Since the extinction cross-section did not change between fresh and dark aged BBA of the same aerosol, it can be inferred that there was no change in the SSA during dark aging of

flaming-dominated aerosol. This could potentially be due to limited emission of the nighttime oxidants and VOCs, unlike during smoldering-dominated combustion.

**3.5 Impact of photochemical aging**

To study the impact of photochemical aging on the optical properties of aerosol, we performed the aging of BB aerosol with the UV lights turned on. In addition, to simulate the impact of photochemical aging in a polluted environment, we added VOCs (benzene, toluene, and xylene) to mimic urban pollution as described in section 2.3.2. For both conditions, scattering and extinction coefficients were measured after 12 hours of aging in the chamber. Figure 7 shows the comparison of SSA from fresh and photochemically aged aerosol. As expected, there was an enhancement

in the SSA of photochemically aged aerosol. A key point to mention is that fresh and aged SSA were from two different burns and we confirmed that under the same burn conditions the SSA of the same fuels remain within the measurement uncertainties of our instruments. Since this study used size-selected aerosol, the increase in SSA was only possible if the particles became less absorbing because of aging, which could potentially be due to the formation of non-absorbing secondary organic aerosol during photochemical aging. An increase in SSA is possible during photochemical aging

due to degradation in brown carbon absorptivity (Sumlin et al., 2017) but the impact of brown carbon on SSA in mid-visible wavelength is small (Pokhrel et al., 2016). BrC components undergo photochemical transformations during atmospheric transport, including photobleaching or photoenhancement of their absorption coefficients. For example, the field studies of Forrister et al. (2015) and Selimovic et al. (2018) observed a substantial decay in aerosol UV light absorption in biomass burning plumes corresponding to a half-life of 9 to 15 hours. Recent laboratory and field studies

suggested that OH oxidation in the atmosphere may alter the optical properties of BrC, leading to absorption enhancement or bleaching (Schnitzler and Abbatt, 2018;Sumlin et al., 2017;Dasari et al., 2019). These studies were made at wavelengths of 375 and 405 nm, while ours were done in the visible range, so they are not directly applicable to this work. As evident from Figure 7, after 12 hours of aging, the BB aerosol becomes highly scattering, leading to SSA values of greater than 0.9 in the mid-visible wavelengths, even though fresh BB aerosol were highly absorbing,

with SSA below 0.8.

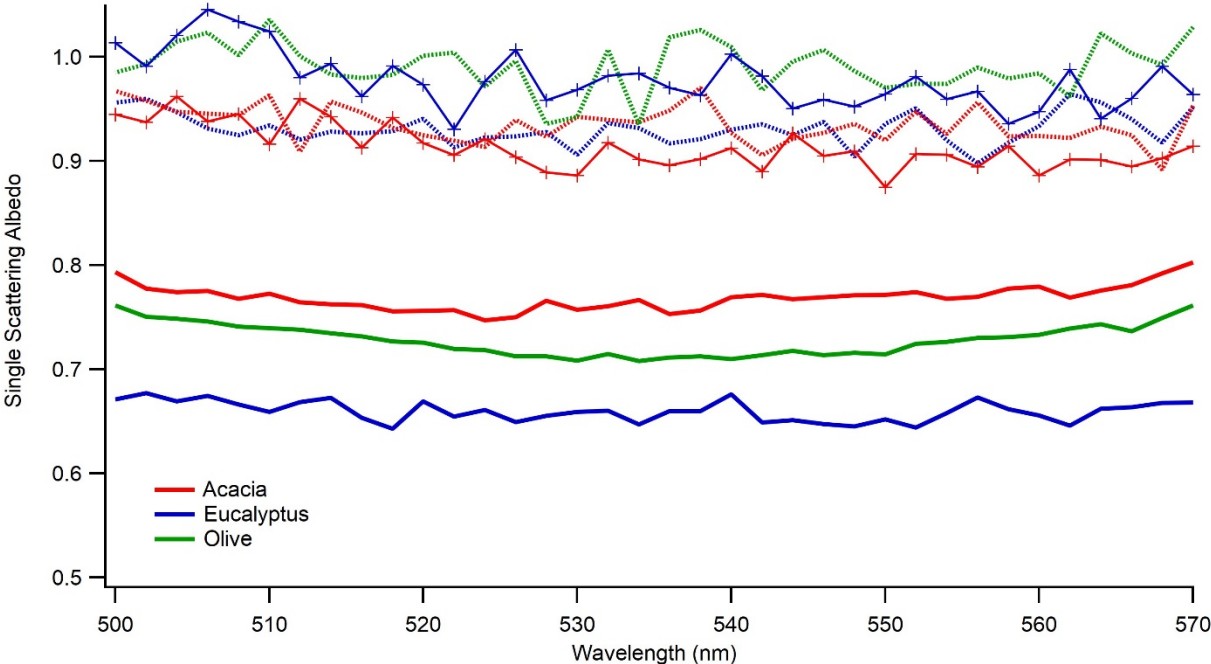

**Figure 7**: Impact of photochemical aging (light and light plus VOC) on SSA of 300 nm size selected smoldering-dominated aerosol. Solid lines are for freshly emitted particles, dotted lines are for aging under the light, and solid lines with symbols are for aging under light plus VOCs. Different colors are for the different fuels listed in the legend.

Despite the use of anthropogenic VOCs with concentrations larger than those average values found in urban and suburban regions of South Africa, no distinct effect was observed for SSA values of BB aerosol produced during smoldering-dominated combustion. This is most likely because we took our measurements after 12 hours of aging, which was enough time for the scattering nature of the SOA produced during aging to drive measured SSA values to unity, regardless of the chemical pathway taken due to the addition of VOCs. While it is possible that the relatively long aging time could obscure some of the effects due to the presence of VOCs, it is also possible that molecular species are dominated by combustion products, and the effects of additional VOCs are insignificant. The later would suggest that the three-aromatic species representing anthropogenic pollutants do not seem to affect the optical properties of BB aerosol. Another possibility would be that high NOx concentrations would prevent the formation of ozone, which would hinder SOA formation from aromatic species. Indeed, examining the effects of aging on the chemical composition of BB aerosol (Figure S9 and Tables S3 and S4) shows very few species that could be attributed to anthropogenic VOCs; specifically, only dihydroxyphthalic acid produced from xylene. For both Eucalyptus and Acacia, an isomer of dihydroxybenzene, such as resorcinol or catechol, was removed to the highest degree from the fresh BB aerosol upon photochemical aging. Generally, very few compounds were produced to a significant extent, and both fuels were dominated by a loss of chromatophoric lignin pyrolysis and distillation products. This includes a great number of functionalized benzaldehyde, benzoic acid, and coumarin species. Not surprisingly, the associated absorbance from these chromophores, mostly from 200 – 350 nm, was also attenuated with respect to age (Figure S10). This may be caused in part by the photo-bleaching effect created by the irradiation of UV light for 12 hours,

heterogeneous OH oxidation, and SOA formation of non-chromophores. It is possible that molecular absorption extends to the mid visible region of the spectrum, as evidenced by its negative and non-zero slope of absorption vs. wavelength in the 500-570 nm region. This corresponded well with absorption cross section measurements of 300 nm

particles and the comparison holds when the comparison was made in terms of an Ångstrom absorption exponent. Additionally, photochemical aging produced more absorption at longer wavelengths, which resulted in a negative Ångstrom absorption exponent and a positive slope in this region. This could be due to the production or persistence of compounds that absorb at such long wavelengths, such as nitro-aromatic species, where one isomer that was either nitroanisole, nitrocresol, or possibly nitrobenzyl alcohol was observed in mass spectra, and functionalized PAHs,

where dihydroxyanthracene was observed to be produced; likely from the oxidation of anthracene.

This fact suggests that a study with higher temporal resolution is needed to simulate the impact of the VOCs on aerosol SSA, where continuous or much more frequent measurements are needed to determine the impact of urban pollution on aerosol SSA. Such a study, using continuous measurements, was not possible for our setup when particles were also size selected. Like dark aged conditions, we were not able to estimate SSA for flaming-dominated

combustion due to the low particle concentration and highly absorbing nature of the aerosol. A chemical analysis of aged BB aerosol produced at this temperature revealed very few changes, suggesting that only a few molecular species were produced by combustion at this temperature.

During aging, the size distribution of the particles was measured every 5 minutes using an SMPS. To account for the impact of additional VOCs on secondary aerosol formation, we estimated the OA enhancement from the time

series of an SMPS spectra during these experiments. This was done by applying a fixed wall loss rate constant as estimated in our previous study (Smith et al., 2019) and assuming a similar loss rate for POA vs SOA. We also assumed that OA is the major aerosol fraction emitted during smoldering-dominated combustion. In addition, we made an assumption of constant density during aging, which gave us a lower estimate of the OA enhancement because the density of aerosol increases with age compared to POA (Tkacik et al., 2017). The OA enhancement calculation method

used in this study was based on the work by Saleh et al. (2013). Briefly, we applied the wall loss time constant to estimate the potential decrease in the OA mass loading only due to wall loss based on the OA mass before the lights were turned on. The OA enhancement factor was estimated by taking the ratio of measured aerosol mass to the predicted aerosol mass based on the wall loss rate constant. It was shown that partitioning of vapors to walls in laboratory experiments may alter apparent SOA production (Hodshire et al., 2019). For the particle wall loss rates

used in this work, we did not correct for partitioning of vapors to walls, which was not generally negligible (Krechmer et al., 2016). Furthermore, it was shown that the tubing between the tube furnace and smog chamber might create loss in the gas phase precursors for SOA formation (Pagonis et al., 2017;Deming et al., 2019;Liu et al., 2019). Relatively greater losses of particles with increased length of sampling tubes was observed (Kumar et al., 2008). Our connecting tubing was short enough (0.5 m) to neglect such a loss. Figure S11 shows the time series of OA enhancement during

different aging conditions. We did not observe a distinct difference between the OA enhancement factors under light aged and light plus VOC aged cases. This could potentially be due to our assumptions and the uncertainty related to the SMPS where the uncertainty in the particle count by the WCPC is ±10%. Previous field and modeling studies found significant enhancement in the SOA formation due to impact of urban pollution (Shrivastava et al., 2019). This

fact also suggests that we need more rigorous study to simulate impact of urban pollution of secondary aerosol formation in the laboratory.

Although as an ACSM was not available early in this study, we designed an experiment to compare the performance of the OA enhancement calculation based on the SMPS with the ACSM during a separate burn. As mentioned earlier, we used a first order decay of the POA based on the estimated wall loss rate constant from the chamber characterization experiments. Figure S12 shows the comparison of OA enhancement using the ACSM OA mass loading vs the estimated submicron aerosol mass based on the SMPS. In general, the trend of OA enhancement was similar but the SMPS seems to underestimate the actual OA enhancement compared to the ACSM. The difference between the OA enhancement estimated by ACSM and SMPS was within 10%, indicating that estimated OA enhancement lies within the SMPS uncertainty range.

## 4 Conclusions

Biomass burning is the major source of atmospheric primary particles and vapors, which are precursors for secondary aerosol. BB aerosol have been extensively studied through both field and laboratory environments for North American fuels to understand the changes in optical and chemical properties as a function of aging. There is a clear research need for a wider sampling of fuels from different regions of the world for laboratory studies. This work is such an attempt to study the optical and chemical properties of fuels common in east Africa and represents the first such study.

The existence of significant variability in the observed field and laboratory measurements has been reviewed recently (Hodshire et al., 2019). While laboratory studies provide control over environmental and chemical conditions to study aging by controlling one variable at a time, it may not necessarily recreate atmospheric conditions in atmospheric plumes in the field. Differences in fuel mixture and fuel conditions, such as moisture content, can lead to different emissions. There is a difference in dilution rates between field studies, which is variable, and laboratory studies, which are not variable. Other differences include temperature differences, background OA concentrations, and wall losses. Despite the gap in reconciling field and laboratory studies, some limited comparisons can be made.

For fresh emissions, SSA showed no pronounced size dependence for flaming-dominated combustion, whereas same size dependence was observed for smoldering-dominated combustion. This may be due to the impact of multiply charged aerosol, which is not discriminated by the DMA. For the wavelength range used in this study, no wavelength dependence of SSA was observed under all conditions. However, SSA showed dependence on fuel type in general, even under the same combustion conditions.

In general, combustion temperature plays a major role in the optical properties of the emitted aerosol. In all cases the measured SSA values for flaming-dominated combustion were in the range between 0.287 and 0.439, indicating highly absorbing aerosol, which corresponds to aerosol dominated by black carbon. We observed a large increase in the SSA during smoldering-dominated combustion, which was in the range between 0.66 and 0.769. Under the same combustion conditions and airflow, there was a clear dependence of SSA on fuel type, with eucalyptus

producing aerosol with higher SSA values than olive and acacia. However, these variations are relatively small, indicating that the SSA was more controlled by the combustion conditions than the fuel type.

Negative mode UPLC/DAD-ESI-HR-QTOFMS analyses of fresh BBA from flaming-dominated combustion suggest that there was some chemical constituent that was not being captured by this analysis. Given that Eucalyptus has a higher SSA than Acacia for the flaming-dominated combustion this would indicate that Eucalyptus has more

non-absorbing OA that was not observed by chemical analysis, such as the presence of eucalyptol. Smoldering-dominate combustion of Eucalyptus and Acacia produced a variety of compounds in common, such as lignin pyrolysis products, distillation products, and cellulose breakdown products. Several lignin pyrolysis products and distillation products were more prevalent in Eucalyptus than in Acacia, while pyrolysis products of cellulose and at least one nitroaromatic species were more prevalent in Acacia. This is consistent with the higher SSA of Acacia when compared

to Eucalyptus for the smoldering-dominated combustion, since lignin pyrolysis products and distillation products are known chromophores, while compounds derived from sugars and cellulose are non-chromophores.

Regardless of fuel types, there occurred an increase in SSA during dark aging, with some fuel dependence, the largest of which was observed for smoldering-dominated aerosol emitted from olive. A significant increase in the scattering and extinction cross-section (mostly dominated by scattering) was observed, indicating the occurrence of

chemistry, even during dark aging. Based on the relevant literature, we hypothesize that nitrogen-containing secondary organic aerosol is formed during dark aging (Li et al., 2015;Hartikainen et al., 2018;Nguyen et al., 2011;Tiitta et al., 2016) . It is possible that this SOA is absorbing in this region, which was offsetting the effect of an increasing OA fraction with age. A chemical analysis of dark aged aerosol during these experiments is planned for future work, and should allow us to test this theory.

After 12 hours of photochemical aging, BB aerosol becomes highly scattering with SSA values above 0.9, even though fresh emissions were more absorbing with SSA below 0.8. This can be attributed to oxidation in the chamber.

Due to the very low number concertation of flaming-dominated aerosol, the results were inconclusive, and we plan to conduct measurements by increasing the amount of fuel burned. We also attempted to simulate polluted

urban environments by injecting anthropogenic VOCs into the chamber, but no distinct difference was observed. An examination of the chemical composition of aged BBA shows very few species that could be attributed to anthropogenic VOCs. Generally, very few compounds were produced to a significant extent, and both fuels were dominated by a loss of chromatophoric lignin pyrolysis and distillation products, likely caused by a combination of photo-bleaching, heterogeneous OH oxidation, and SOA formation of non-chromophores. No significant OA

enhancement was observed because of the VOC injection either, even though significantly enhanced SOA formation was observed in polluted environments (Shrivastava et al., 2019;Shrivastava et al., 2015). This suggests a need for more rigorous controlled time dependent measurements.

To our knowledge, this is the first laboratory study of optical properties of east African biomass fuels for domestic use. Ongoing work includes systematic study of optical properties using six different African fuels as a

function of aging, burn conditions, VOC concentration, and RH and morphology.

**Author contribution:** Damon Smith conducted the experiments and analyzed the data; Marc Fiddler and Solomon Bililign designed the experiments and contributed to writing and editing. Rudra Pokhrel contributed to the data analysis and interpretation.


**Competing interests:** The authors declare that they have no conflict of interest.

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
