# Peer review of "Laboratory studies of fresh and aged biomass burning aerosol emitted from east African biomass fuels-PART 1-Optical properties"

_Atmospheric Chemistry and Physics, 2019_

## Referee Comment (RC1) · Anonymous Referee #1 · 13 Mar 2020

The paper summarizes results on optical properties of aerosols generated by burning of 3 common fuels in Africa. Some measurements were carried out on fresh BBOA, some on BBOA aged in the dark, some on photooxidized BBOA in the presence or absence of additional VOCs (typical urban aromatics). Fuels were burned at low (500 C) and high (800 C) temperatures. Non-refractory chemical composition of the aerosol was also measured, although not really discussed here. Single scattering albedo data, determined on size-selected aerosols, at mid-visible confirmed production of more absorbing aerosols under high temperatures. There was not a strong wavelength dependence to SSA between 500-570 nm. Dark aging of the BBOA resulted in increase in SSA possibly due to condensation of SOA and increase in the scattering cross section. The novelty of the experiment is in the selection of the fuels. However, I see some flaws in the approach (contribution of multiply charged particles was not corrected for in Mie calculations); the paper does not seem complete without the description of the compositional measurements (some conclusions are drawn on the composition of SOA without providing any support; there were no measurements of BC, yet overall refractive indices were estimated to use in Mie calculations); last the chamber oxidation experiments in the presence of additional VOCs don't seem to have worked. There were also several sentences that were confusing and need to be rephrased/clarified. Overall I don't find the quality of this paper appropriate for ACP and cannot support any revisions. Other specific comments are listed below: - L 20-21: Bad grammar - L 23-24: bad grammar - L32: "injecting" and not "ejecting" - L33: it's unclear why measurements after 12 hrs of injecting VOCs and BB aerosol in the chamber were not useful. - L 43: "measuring the…" - L47: "partial evaporation…" - L85: "authors'…" - L112: The sentence related to drying of the fuels should be moved earlier, before the weighing discussion. - L115: the EF mentioned is for tropical forests, which is not really representative of the African fuels studies here. Please provide your justification for using this value. - L164-165: It's unclear what is actually meant here "SSA and AAE were derived from these relationships using observations of CO, CO2, OA, and BC." - L187: Temperature change even if very small should be included since it can impact gas-aerosol partitioning and evaporation of BB-POA. - L216-217: Consider indicating what the aerosol number concentrations, and absorption and scattering coefficients typically were after 24-hr flushing and before start of a new expt. - L221: what does it mean that "The experiments were repeated after keeping the BB aerosol in the chamber overnight (24 hours) without the UV lights"? - L231: indicate manufacturer and level of purity - L245-251: Typically urban concentrations of VOCs are expressed as volumetric mixing ratios (ppmv or ppbv, etc). Is that not the case for the aromatics indicated here? If not, to clarify, you need to include ppmm or ppm by mass. If indeed the mixing ratios of 5:14:6 was ppmv based, then one expects different volumes (and also different masses of them based on their density) of each to be injected and the

calculation here is not correct. - L254-256: what VOC mixing ratios were achieved in the bag so readers can compare them with the typical urban mixing ratios of these VOCs in Africa? - L267: what does "710 $\mu$m impactor inlet" mean? It seems 710 um is not related to the size cut. - L269- 270: how was the concentration of multiply charged particles accounted for? Those can significantly impact the measured optical coeffi-cients given their larger physical size, even if their number concentration is not high. This is mentioned in lines 318-320, but still no consideration is given for correcting 200 nm particle concentrations. - Equation 2: need to define tau and tau0, sigma_ext and N(CRD). - L294: Since the CPC is placed after the optical instrument, what is the frac-tionall loss of particles in the CRD and nephelometer? - L316: it is worth indicating what each of these uncertainties are. - L330-331: It's unclear what the CE value was and how it was determined for this one species. What assumptions need to be made to justify using the same CE for all the species? - L340: this sentence needs to be rephrased "The optical properties were also measured as a function of different forms of aging: dark aged, photochemically and photochemically aged with added VOC's lights on in the presence of VOCs injected into the chamber before particles were in-troduced at both temperatures." - L351 and 363: section 2.3.1 is not in the paper - L357: How was concentration of BC taken into account to calculate the SSA? And how were the aerosols treated? Internal or external mixtures? These details need to be fully explained. - L361-362: why would the impact of multiply charged particles not be present in lower combustion temperature samples? How different were the chamber size distributions under these conditions? The geometric mean for all fresh aerosols seems to be the same and 50 nm (Fig 1). - Figure 3. The legend needs to include the imaginary number indicator "i" - L380: Do the dashed lines show the uncertainties in SSA or the variability of the measurements, e.g., standard deviation of the average? Please be specific. - L391-392: Unlike what's mentioned here, SSA for eucalyptus aerosol is not uniformly higher or lower than other fuels; please correct this statement. This needs to be corrected in the conclusions as well. - L382-400: The discussion on burn temperature and BC vs BrC emissions gets repeated; consider describing this

dependence more concisely and to the point only once. - Figure 5: why is the SSA for 400 nm included here whereas in other plots 300 nm observations were included? What is the reason for having much larger error bar on one of the data points? - L414: The sentence related to the higher uncertainty needs to be rephrased. - L419-421: I don't think the example with actual values of SSA are needed. Also the last sentence is stating something that has been known in the community for a long time; therefore ,it's not worth reiterating or at least provide proper literature reference. - L431: Why is particle dynamic expected to be different at night compared to daytime such that it would influence nighttime oxidation differently? Please clarify. - L461: there's no basis for suggesting nitrogen-containing OA as opposed to other types of SOA were formed under these dark aging conditions since no information on composition was provided. Please rephrase/remove this in the Conclusions as well. - L487: Section 2.2.1 includes description of photooxidation without additional VOCs - L505-507: the explanation doesn't seem to be valid. Why can't it be that the SOA form these VOCs has the same optical characteristic as the SOA formed in the absence of the VOCs? L539: it actually appears that there was insignificant additional SOA formed from oxidation of aromatics that were added to the chamber. What was the NOx level in the experiments? Perhaps the high NOx conditions of the burns lead to low SOA yield from these precursors and therefore no significant SOA is observed. If results from aging in the presence of additional VOC were not conclusive, I suggest removing all the discussion related to it throughout the paper. - L 516-517: I'm not following why continuous size-selection was not possible during these aging experiments.

---

## Referee Comment (RC2) · Anonymous Referee #2 · 17 Mar 2020

This manuscript presents measurements of single scattering albedo (SSA) of size-selected aerosols emitted from controlled combustion of African biomass fuels under three conditions: fresh emissions, dark aged aerosols and photo-chemically aged aerosols. Three types of wood fuels were combusted in a tube furnace at two different temperatures (500 C and 800 C) and an indoor smog chamber was used to age aerosols in clean and polluted (VOC rich) environments. The authors claim that the significance of their work lies in providing optical and chemical characterization of a previous unstudied group of fuels that contribute significantly to aerosol emissions in Africa. However, there are no novel findings reported in this study and claims of significance are greatly overstated. While the particular fuels in this study might not have been

characterized, there is a robust body of literature regarding the effect of combustion conditions on optical properties of emitted aerosols in controlled (example: Chen and Bond, 2010, ACP; Saleh et al., 2018, ES&T) as well as representative household use (Roden et al., 2006, ES&T; Chen et al., 2012, ES&T) settings. This study was limited by a lack of chemical characterization and SSA measurements limited to mid-visible wavelengths, and therefore could only reiterate the well-known effect of combustion temperature on absorption efficiency. The aging experiments show that both dark and photochemical aging reduce the absorption efficiency of size-selected aerosols (photochemical more so than dark) but no chemical properties were measured to illuminate the mechanism of absorption loss. Further, the aging results are only presented for 500 C aerosols because (Line 471): "Therefore, due to the very low number concentration and highly absorbing nature of the particles, the scattering coefficient at 800 C was below the detection limit of our nephelometer during the aging experiments." The authors propose that future studies will include these missing measurements (performed by increasing the amount of fuel burnt) but I am puzzled why these changes were not made for this study. There are similar problems with aging experiments in a polluted environment (Line 505: "This is because we took our measurements after 12 hours of aging, which seems long enough to characterize the impact of the added VOC due to aging in UV. This fact suggests that a more carefully controlled study is needed to accurately simulate the impact of urban pollution on aerosol single scattering albedo") that indicate that the authors did not rigorously handle their motivating hypotheses, leaving glaring holes in their manuscript.

Aside from concerns about significance and study design, there are significant issues in how the manuscript is presented. Instances of grammatical errors and confusing sentence construction are far too many to enumerate but more importantly, several arguments/claims are not supported by findings in this study or citations from literature. The authors establish that the fuels studied here are household fuels and acknowledge the potential differences between typical household use and controlled burning. They do not present any discussion of how findings from controlled combustion can be extended to a more realistic condition: this undermines the purported importance of their findings. Further, they designate their 800 C burn condition as flaming (a reasonable assumption) and 500 C burn as smoldering (which is much higher than smoldering temperatures in literature). These assumptions are not substantiated with any further evidence. The SSA values reported for smoldering appear too low for pure smoldering combustion (eg. - those in Sumlin et al., 2018, JQSRT) and I am not convinced that the authors ensured that they are not from mixed combustion conditions. Line 415 (comparing SSA values here with previous studies) states: "This could explain why our SSA calculations for BrC was lower than expected". All measurements in this study are for total aerosols, BrC is mentioned without any justification. Many hypotheses are presented for the aging observations however the study was not conducted in a way that allows any plausible claims about "night-time formation or aromatic nitrogen containing compounds", for example. SOA formation is presented as a hypothesis for SSA reduction (Line 492) during photochemical aging but fragmentation of absorbing aerosols is not considered. Finally, the choice of figure type for representing the results in figures 4, 6 and 7 is baffling to me: why are SSA values plotted over this very narrow range of wavelengths? Clearly, no wavelength dependence can be seen between 500 and 570 nm. I fail to see the purpose of multiple figures that contain a series of zigzagging flat lines.

Overall, a lot of more thought is needed in designing experiments and choosing the type of measurements needed to answer important questions about the chemical and optical properties of African biomass fuels. A lot more care in presenting those findings and placing them in context of recent studies is also required.

---

## Author Comment (AC1) · 15 Apr 2020

The full response is attached as a pdf file with figures. RESPONSES to REFEREE #1

The paper summarizes results on optical properties of aerosols generated by burning of 3 common fuels in Africa. Some measurements were carried out on fresh BBOA, some on BBOA aged in the dark, some on photooxidized BBOA in the presence or absence of additional VOCs (typical urban aromatics). Fuels were burned at low (500 C) and high (800 C) temperatures. Non-refractory chemical composition of the aerosol was also measured, although not really discussed here. Single scattering albedo data, determined on size-selected aerosols, at mid-visible confirmed production of more absorbing aerosols under high temperatures. There was not a strong wavelength dependence to SSA between 500-570 nm. Dark aging of the BBOA resulted in increase in SSA possibly due to condensation of SOA and increase in the scattering cross section. The novelty of the experiment is in the selection of the fuels. However, I see some flaws in the approach (contribution of multiply charged particles was not corrected for in Mie calculations); the paper does not seem complete without the description of the compositional measurements (some conclusions are drawn on the composition of SOA without providing any support; there were no measurements of BC, yet overall refractive indices were estimated to use in Mie calculations); last the chamber oxidation experiments in the presence of additional VOCs don't seem to have worked. There were also several sentences that were confusing and need to be rephrased/clarified. Overall, I don't find the quality of this paper appropriate for ACP and cannot support any revisions.

AUTHORS RESPONSE: We thank the reviewer for valuable and helpful suggestions. Embarrassing grammatical errors and sentences that seemed confusing will be corrected in the revision and changes made to the manuscript are provided here. Detailed responses are provided on the scientific questions raised. The authors feel that the reviewer either misunderstood our descriptions and overlooked some aspects of the paper. We will show this is indeed true in our detailed responses with relevant references to each question and comment. We feel that there is rush to dismiss and devalue the work and undermine the paper instead of giving us a chance to respond and clarify our claims. We hope our responses and explanations will convince the referee to make a different final determination. Three main issues raised by the reviewer in his/her the first paragraph is discussed below:

A. "Non-refractory chemical composition of the aerosol was also measured, although not really discussed here"

We would like to point out to the reviewer that this work is presented and submitted as a two-part paper, where part 1 was focused on optical properties and part II focused

on chemical composition and characterization published in ACPD (Smith et al., 2020). Further integration of the two manuscripts will be done, and summaries from Part 2 will be incorporated into the revised manuscript. Following the paragraph contrasting eucalyptus and acacia combusted at 800o C, the following text will be added to line 393:

"In the companion paper to this (Part 2), methanol extracts from BBA collected on Teflon filters were analyzed by ultra-performance liquid chromatography interfaced to both a diode array detector and an electrospray ionization high-resolution quadrupole time-of-flight mass spectrometer (UPLC/DAD-ESI-HR-QTOFMS) in negative ion mode. This was used to determine the relative abundance and light-absorption properties of biomass burning organic aerosol constituents. MS analysis of BBA extracts from combustion at 800 °C revealed very little difference between the two fuel types, suggesting that there are either very few BrC species produced for either fuel under these combustion conditions, or there are numerous species that are essentially the same between the samples. However, given that Eucalyptus has a higher SSA than Acacia, this would suggest that Eucalyptus has more non-absorbing OA, or at least less absorbing than BC. Since it is Acacia that appears to have many more low-abundant organic constituents, several possibilities exist to explain these differences in SSA, as explored in more depth in Part 2. It is likely that Eucalyptus combustion products are not captured by some aspect of the extraction and UPLC/DAD-ESI-HR-QTOFMS analyses, that the observed differences in SSA are due to morphology differences, or some combination thereof. One potential explanation would be the presence of significant amounts of eucalyptol in the BBA, which is a large fraction of Eucalyptus oil, and is a cyclic ether that lacks any basic functionality amiable for negative ion mode analysis, has good solubility in alcohols, and does not absorb in the UV and visible. An examination of the UV-Visible spectra from the DAD shows no absorbing species in either region."

The following paragraph will be included in manuscript following the discussion of fresh emissions produced by combusted at 500o C, added to line 410:

"Chemical analysis revealed that, when combusted at 500 °C, eucalyptus and acacia had a variety of compounds in common, such as lignin pyrolysis products, distillation products, and cellulose breakdown products. Several lignin pyrolysis products and distillation products are more prevalent in Eucalyptus than Acacia, while pyrolysis products of cellulose and at least one nitroaromatic species were more prevalent in Acacia. Given that these lignin pyrolysis and distillation products are known chromophores and are more prevalent in Eucalyptus than Acacia, while Acacia has a higher abundance of non-chromophores derived from sugars and cellulose, one would assume that Eucalyptus would be more absorbing in the visible (i.e. have a lower SSA) than Acacia. Despite the chemical analysis not capturing absolute amounts of OA, Acacia was found to have an SSA that is higher than Eucalyptus by 0.1 to 0.2, which is consistent with chemical measurements. This suggests that Acacia has either larger absolute amounts of non-chromophore compounds or Eucalyptus has a greater quantity of chromophores whose absorptive properties extend to the 500 – 570 nm region of the visible spectrum. An analysis of the chromatographically-integrated UV/Visible spectrum shows that there are chromophores whose absorption features peak ∼290 nm and extend into the 500 – 570 nm region, though a normalized spectrum does not appear to show drastic differences between species."

Later, starting at line 504 with the sentence "We attempted...", the remainder of this paragraph and the next (after Figure 7) will be replaced with the following text:

"Despite the use of anthropogenic VOCs a concentrations larger than those average values found in urban and suburban regions of South Africa, no distinct effect was observed for SSA values of BBA produced during combustion at 500 °C. While it is possible that the relatively long aging time could obscure some of the effects due to the presence of VOCs, it is also possible that combustion products dominate molecular species and the effects of additional VOCs are insignificant. The later would suggest that anthropogenic pollution does not seem to affect the optical properties of BBA. Indeed, in examining the effects of aging on the chemical composition of BBA shows very

few species that could be attributed to anthropogenic VOCs; specifically, only dihydroxyphthalic acid produced from xylene. For both Eucalyptus and Acacia, an isomer of dihydroxybenzene, such as resorcinol or catechol, was removed to the highest degree from the fresh BB aerosol upon photochemical aging. Generally, very few compounds were produced to a significant extent and both fuels were dominated by loss of chromatophoric lignin pyrolysis and distillation products. Not surprisingly, the associated absorbance from these chromophores, mostly from 200 – 350 nm, also attenuated with respect to age. This may be caused in part by the photo-bleaching effect created by irradiation of UV light for 12 hours, heterogeneous OH oxidation, and SOA formation of non-chromophores. This fact suggests that a study with a higher temporal resolution is needed to simulate the impact of the VOCs on aerosol SSA, where continuous or much more frequent measurement are needed to determine impact of urban pollution on aerosol single scattering albedo. Such a study, using continuous measurements, is not possible for our setup when particles are also size selected. Like dark aged conditions, we were not able to estimate SSA for combustion at 800 oC due to the low particle concentration and highly absorbing nature of the aerosol. A chemical analysis of aged BBA produced at this temperature revealed very few changes, suggesting there are few molecular species produced by combustion at this temperature."

B. "However, I see some flaws in the approach (contribution of multiply charged particles was not corrected for in Mie calculations"

There seems to be a misunderstanding regarding Mie calculation. As stated on P 11, L 356-358, we estimated the SSA of size-selected aerosol using the refractive indices estimated by Bond and Bergstrom, (2006) and Levin et al. (2010) for BC and bulk aerosol. The whole purpose of this effort was to explore if our estimated SSA for 200 nm size particles was impacted by the presence of multiply charge particles. The impact of multiply charged particles on SSA for 300 and 400 nm size particles was minimal. We only run the Mie models to make a qualitative comparison of Mie result with our measurements, focusing on the SSA dependence on particle size. The

disclaimer "Neither line is a fit to the data." Will be added to the caption of Figure 3. Further discussion on multiply charged particles is provided below in response to reviewer comments on L361-362.

C. "Some conclusions are drawn on the composition of SOA without providing any support"

The focus of the work is not SOA formation. However, as we observe changes in SSA we looked at the data more closely as stated on P 16, Line 460, we hypothesized that there could be SOA production during dark aging, and we cited previous work showing nitrogen-containing SOA production under nighttime conditions as potential chemical basis for this observation. As clearly stated in our discussion and in the abstract section, we hypothesized SOA production during dark aging but never definitively concluded about composition of SOA, as we do not have measurements for these experiments. We were inferring to work that was previously reported and focused on SOA formation during dark aging. (Tiitta et al., 2016;Li et al., 2015;Hartikainen et al., 2018). It seems that our statement regarding this was misunderstood and needs further clarification, which will be included in the revised draft. Furthermore, the statement "there were no measurements of BC, yet overall refractive indices were estimated to use in Mie calculations" is not valid in our work given we never stated that we estimated refractive indices nor do we present any refractive indices of our own at any point. We have provided more details about this in response to the comment on L357. The reviewer is mischaracterizing things as "additional VOCs don't seem to have worked". Just because something didn't significantly change, doesn't mean it didn't work. It just means that it doesn't seem to have had an impact.

Other specific comments are listed below: - L 20-21: Bad grammar – Agreed. It will now read as "This work represents the first such study of the optical and chemical properties of three wood fuel samples used commonly for domestic use in east Africa." L 23-24: bad grammar – Agreed: "values are in the range between 0.287 and 0.439 while the SSA for fuels combusted at 500o C, the range is between 0.66 and 0.769"

Interactive
comment

L32: "injecting" and not "ejecting" – Agreed: ejecting will be replaced by injecting L33: it's unclear why measurements after 12 hrs of injecting VOCs and BB aerosol in the chamber were not useful. – We believe that measurements should have been made soon after injecting the VOC's L 43: "measuring the. . ." – "of" will be removed L47: "partial evaporation. . ." – Agreed: partial evolution should be partial evaporation L85: "authors'. . ." – Will be replaced with "To our knowledge"

L112: The sentence related to drying of the fuels should be moved earlier, before the weighing discussion. –

AUTHORS RESPONSE: We agree with the reviewer. Sentence related to drying of the fuels will be moved earlier, before the weighing discussion.

L115: the EF mentioned is for tropical forests, which is not really representative of the African fuel's studies here. Please provide your justification for using this value. –

AUTHORS RESPONSE: We agree with the reviewer. We will delete the whole sentence talking about the EF and will add the following text:

"During all the experiments, we normally burned 0.5 grams of fuel which produced about 600 to 800 $\mu$g m-3 of mass loading in the chamber. The mass loading was estimated by determining the total aerosol volume, based on measuring the volume distribution with an SMPS and assuming a density of 1 g cm-3 for fresh aerosol."

L164-165: It's unclear what is meant here "SSA and AAE were derived from these relationships using observations of CO, CO2, OA, and BC." –

AUTHORS RESPONSE: We agree with the reviewer. We will delete the whole sentence.

L187: Temperature change even if very small should be included since it can impact gas-aerosol partitioning and evaporation of BB-POA. –

AUTHORS RESPONSE: Chamber temperature started at room temperature (around

20 °C or slightly above) and increased to a maximum of 30 °C after 5 hours of use when all the UV lights were turned on, with most of the increase happening within the first hour (Smith et al., 2019). The figure below is the temperature profile for different runs.

Fig1. Temperature profile of the chamber. Runs 1b and 4b were done on the same day as runs 1a and 4a, respectively, with initial temperatures higher for 1b and 4b than 1a and 4a.

L216-217: Consider indicating what the aerosol number concentrations, and absorption and scattering coefficients typically were after 24-hr flushing and before start of a new expt. –

AUTHORS RESPONSE: Number concentrations were measured and must be below threshold values before a new experiment could begin. These values were typically around 25 – 40 particles cm-3 as measured by the CPC. Absorption and scattering at these low number concentrations were not measured and gives a typical background mass loading of 1ïA■gm-3.

L221: what does it mean that "The experiments were repeated after keeping the BB aerosol in the chamber overnight (24 hours) without the UV lights"? –

AUTHORS RESPONSE: We are sorry for the typo. Now the sentence will read "For dark aging, measurements were repeated after 12 hours without the UV lights".

L231: indicate manufacturer and level of purity –

AUTHORS RESPONSE: These mixtures were composed of benzene (≥99.9%, Sigma-Aldrich), toluene (99.99%, Acros Organics), and ortho-xylene (99%, Alfa Aersar). This information will be added in the revised manuscript.

L245-251: Typically, urban concentrations of VOCs are expressed as volumetric mixing ratios (ppmv or ppbv, etc). Is that not the case for the aromatics indicated here? If not, to clarify, you need to include ppmm or ppm by mass. If indeed the mixing ratios of

5:14:6 was ppmv based, then one expects different volumes (and different masses of them based on their density) of each to be injected and the calculation here is not correct. –

AUTHORS RESPONSE: The reviewer is correct. There seems to have been some miscommunication among investigators during the experimental design of this particular experiment. See our response to the next comment, where both will be addressed.

L254-256: what VOC mixing ratios were achieved in the bag so readers can compare them with the typical urban mixing ratios of these VOCs in Africa? –

AUTHORS RESPONSE: The two paragraphs and the first table, from lines 239 – 263, will be replaced with the following text:

"To represent a polluted urban environment, we used emission inventory for urban environments from South Africa. This does not necessarily represent the east African emission inventory, but this serves as a baseline, since this is the only available data to us for the continent. This was obtained from South African Air Quality Information System (SAAQIS) and included concentrations of NOx, NO, NO2, CO, O3, benzene, toluene, ortho-xylene, and ethylbenzene for several South African Sites (Diepkloof, Kliprivier, Three Rivers, Sharpeville, Sebokeng, Zamdela, Thabazimbi, Lephalale, Phalaborwa, and Mokopane). The VOC data was obtained from the two weeks (M-F) of July 11 – 15 and July 18 – 22, which was in the middle of the peak burning season for South Africa for the year 2016. The urban areas (Diepkloof and Kliprivier) had combined average mixing ratios of 1.16, 3.48, and 1.44 ppbv for benzene, toluene, and o-xylene, respectively. Suburban areas (Three Rivers, Sebokeng, and Zamdela) had combined average mixing ratios of 1.69, 4.02, and 0.70 ppbv for the aforementioned gases, respectively. Interestingly, suburban regions had somewhat higher average benzene and toluene mixing ratios, though o-xylene was only half the average urban concentration. A mixture was prepared using equal by volumes of benzene, toluene, and o-xylene, and 2.5 mg was injected by syringe into a U-shaped glass tube attached to the chamber. This tube was then flushed by zero air into the chamber. This resulted in a mixing ratio of 29.7, 24.9, and 21.9 ppbv for benzene, toluene, and o-xylene, respectively. The concentration injected into the chamber was approximately 7 – 26 times more concentrated than values found from urban South African emissions and 6 – 18 times more concentrated than suburban values. The reason for these elevated levels was mostly due to sample preparation constraints, since the amounts needed for an exact match were too small for our scale to weigh appropriately. Concentrations in the chamber were intentionally higher than atmospheric conditions, in order to age the BB aerosol faster and accentuate the potential effect of SOA."

L267: what does "710 $\mu$m impactor inlet" mean? It seems 710 um is not related to the size cut. – AUTHORS RESPONSE: That is the diameter of the impactor used right before our DMA. We will rewrite the sentence as "BB aerosol was size selected for optical property measurements by passing the sample through an impactor inlet with a 710 $\mu$m nozzle (3.8 ïĄ■m diameter cut point), charge neutralizer (TSI model 3081), and a long differential mobility analyzer (DMA) (TSI model 3080).

L269- 270: how was the concentration of multiply charged particles accounted for? Those can significantly impact the measured optical coefficients given their larger physical size, even if their number concentration is not high. This is mentioned in lines 318-320, but still no consideration is given for correcting 200 nm particle concentrations.

AUTHORS RESPONSE: As mentioned in the manuscript, concentration of multiply charged particles was not considered in this study. We completely agree with the reviewer that this will significantly impact the measured optical coefficients and cross sections. However, the impact of multiply charge particle on SSA is low (ratio of scattering to extinction coefficients/cross-section). We recently acquired an aerosol particle mass analyzer (APM) and after this comment was posted, we did an experiment using the APM in-line after the DMA and performed optical property calculation for freshly emitted BBA. Detail about the measurement strategy is given by Radney and Zangmeister, 2016). It was found that, due to the presence of multiply charged particles, our

reported SSA values were overestimated by a maximum of 8% for 200 nm &300 nm particles. This information will be included in the revised manuscript.

- Equation 2: need to define tau and tau0, sigma_ext and N(CRD). –

AUTHORS RESPONSE: We thank reviewer for pointing this out. We will define those terms in Equation 2 in the revision.

L294: Since the CPC is placed after the optical instrument, what is the fractionall loss of particles in the CRD and nephelometer? –

AUTHORS RESPONSE: We thank reviewer for pointing this out. We used PSL sphere size standards to estimate the losses in CRDS and Nephelometer. The estimated particle losses in CRDS are 14.2, 14.7, and 11.4 percent for 200, 300, and 400 nm sizes, respectively, whereas for Nephelometer those losses are 8.6, 7.1, and 6.3 percent for those respective sizes. We accounted those losses in our final calculations of extinction and scattering coefficients, as discussed in depth in our previous publication (Singh et al., 2014) and this work is cited in the paper.

L316: it is worth indicating what each of these uncertainties are. –

AUTHORS RESPONSE: We will report those uncertainties. The sentence will now read "The calculation flow determining average extinction cross section ($\sigma$ext), absorption cross section ($\sigma$abs), and single scattering albedo ($\omega$), was already described (Singh et al., 2014)(Singh et al., 2016). The estimated uncertainties are 11%, 15%, and 2.1% for $\sigma$ext, $\sigma$abs, and $\omega$".

L330-331: It's unclear what the CE value was and how it was determined for this one species. What assumptions need to be made to justify using the same CE for all the species? –

AUTHORS RESPONSE: We appreciate reviewer for mentioning this. This sentence will now read as "Since this work does not use mass loading in a quantitative way, we chose a collection efficiency of 1 for all species, similar to a previous study (Ng et al.,

2011)".

L340: this sentence needs to be rephrased "The optical properties were also measured as a function of different forms of aging: dark aged, photochemically and photochemically aged with added VOC's lights on in the presence of VOCs injected into the chamber before particles were introduced at both temperatures."

AUTHORS RESPONSE: We agree with reviewer. We will rephrase the sentence as "The optical properties of dark and photochemically aged aerosol were also measured".

L351 and 363: section 2.3.1 is not in the paper.

AUTHORS RESPONSE: We thank reviewer for pointing out this typo. We will correct that as section 2.4 now.

L357: How was concentration of BC taken into account to calculate the SSA? And how were the aerosols treated? Internal or external mixtures? These details need to be fully explained.

AUTHORS RESPONSE: We think there is a misunderstanding of what we did. We ran Mie model based on the RI of BC from Bond and Bergstrom, (2006) and RI of bulk aerosol from Levin et al, (2010). Based on these model result, we estimated the size selected SSA at 532 nm and did comparison with the SSA calculated from our measurements. The whole purpose of this was to point out two facts: 1. Aerosol formed during 800o C burning are very absorbing (equivalent to pure BC) and 2. Due to the presence of multiply charge particles, our estimated SSA are biased, especially for lower size particles.

L361-362: why would the impact of multiply charged particles not be present in lower combustion temperature samples? How different were the chamber size distributions under these conditions? The geometric mean for all fresh aerosols seems to be the same and 50 nm (Fig 1). - Figure 3. The legend needs to include the imaginary number

indicator "i".

AUTHORS RESPONSE: The typical particle size distribution (corrected for multiple charging) of two different burning temperatures is shown in Figure a. As depicted in the distributions, the higher temperature burn would be more impacted by multiply charged particles because the relative particle concentrations at 200 nm are lower compared to equivalent mobility particles with charges of +2 (314 nm) and +3 (418 nm). On the other hand, for lower temperature burn, concentration at 200 nm size is about 3.5 time larger than that of +2 (314 nm) and about 9 times larger than +3 (418 nm) size. All these facts suggest that there will be more impact due to multiply charged particles for the higher temperature burn than that of lower temperature burn as depicted in Figure a.

Figure 2: Normalized particle size distribution for 500o C and 800o C burning cases.

We will re-plot Figure 1 with x-axis on log scale (shown below as Figure b), which shows the difference in the geometric mean diameter (GMD) of two different burning conditions. For the lower temperature case, it is typically about 85-90 nm whereas for higher temperature burn it is below 40 nm. We will update the legend for Figure 3.

Figure 3: Same as Figure 1 in the manuscript (x axis in log scale).

L380: Do the dashed lines show the uncertainties in SSA or the variability of the measurements, e.g., standard deviation of the average? Please be specific. –

AUTHORS RESPONSE: The dashed lines represent the propagated uncertainty (1 standard deviation) of the SSA, based on extinction and scattering coefficients. The scattering coefficient error is mainly produced by measurement variability from the nephelometer (∼1%), while extinction errors from the CRDS are mainly influenced by variability in the ring-down time. This has been thoroughly described in our system characterization paper (Singh et al., 2014) cited in this paper.

L391-392: Unlike what's mentioned here, SSA for eucalyptus aerosol is not uniformly

higher or lower than other fuels; please correct this statement. This needs to be corrected in the conclusions as well. – L382-400: The discussion on burn temperature and BC vs BrC emissions gets repeated; consider describing this dependence more concisely and to the point only once. -

AUTHOR RESPONSE: We agree with the reviewer. We will rewrite the two paragraphs (L382-409) and it will read as:

"The range of SSA for different fuels combusted at 800o C is 0.287 to 0.439, whereas the range for the same fuels combusted at 500o C is 0.66 to 0.769. The average MCE for 800o C burn cases is 0.974 ± 0.015 and that for 500o C burn cases is 0.878 ± 0.008. These MCE values suggests that 800o C burns are dominated by flaming phase and 500o C burn cases are dominated by the smoldering phase of the burn. The flaming stage of the combustion produces more black carbon and smoldering stage produces more organic carbon (Christian et al., 2003; Ward et al., 1992), which explains the lower SSA values at higher temperature combustion. The impact of combustion temperature on aerosol can also be separated visually by looking at the color of the collected filter samples, as shown in Figure S1. As evident from Figure S1, aerosol emitted from the 800o C combustion looks black, whereas that from 500o C combustion looks brownish, indicating a visual difference between black carbon dominated and organic carbon dominated emissions from same fuel under different combustion temperatures. Although the variation in SSA of the different fuels is dominated by the burning conditions, there occurs a clear but small dependence of SSA on fuel type under the similar burn conditions.

The range of SSA for combustion at 500o C is comparable to previous studies with similar MCE values (Liu et al., 2014; Pokhrel et al., 2016). On comparing the SSA of the three different fuels under two different combustion temperatures, it is apparent that SSA is controlled more by the combustion conditions rather than the fuel types. There is a larger variation in SSA for the same fuel under two different combustion conditions, compared to the variation due to the inter fuel variability under the same combustion

temperature. This result is consistent with a previous study, which showed that SSA is highly correlated with the EC/TC (proxy for the combustion conditions), even for a wide variety of fuels (Pokhrel et al., 2016). A complete list of sizes selected SSA of different fuels measured at two combustion temperatures and under different aging conditions is provided in Table S1."

Figure 5: why is the SSA for 400 nm included here whereas in other plots 300 nm observations were included? What is the reason for having much larger error bar on one of the data points? –

AUTHORS RESPONSE: The reason that this data point seems to be an outlier in terms of its error is because, for that particular experiment, the standard deviation for the ring-down time of the blank was about double what it normally is, even compared to other wavelengths in the same experiment. This propagated to an SSA error of $\pm0.113$. The standard deviation of repeat SSA measurements was only $\pm0.008$, as measurements during this experiment (Olive combusted at 500o C) was extremely reproducible.

L414: The sentence related to the higher uncertainty needs to be rephrased. –

AUTHORS RESPONSE- We thank the reviewer again for pointing out this poorly constructed sentence. It will be rephrased as "This outlier could also be because of the higher error associated with the scattering measured by the nephelometer."

L419-421: I don't think the example with actual values of SSA are needed. Also, the last sentence is stating something that has been known in the community for a long time; therefore, it's not worth reiterating or at least provide proper literature reference. –

AUTHORS RESPONSE: We agree with the reviewer. The sentence with the example is removed and the reference is provided in the last sentence and will read as "This suggests that by simply varying the combustion temperature, we can generate aerosols with very different optical properties and combustion efficiencies (Saleh et al., 2018;Liu

et al., 2014)" in the revision.

L431: Why is particle dynamic expected to be different at night compared to daytime such that it would influence nighttime oxidation differently? Please clarify.

AUTHORS RESPONSE: The particle dynamics is different at night since there is, a pronounced increase of particle size and density which was also observed in previous laboratory and ambient measurements (Reid et al., 1998;Zhang et al., 2011). Even though the RH remains the same in our experiments, it was also shown that RH may facilitate heterogeneous reactions during the night (Li et al., 2015).

L461: there's no basis for suggesting nitrogen-containing OA as opposed to other types of SOA were formed under these dark aging conditions since no information on composition was provided. Please rephrase/remove this in the Conclusions as well. –

AUTHORS RESPONSE: We thank reviewer for pointing this. To make this clear, we are not suggesting nitrogen-containing OA as opposed to other types of SOA. On Line 460 we wrote that "We hypothesized secondary organic aerosol formation as a potential phenomenon happening during dark aging." Then at Line 461 we refer to the previous studies which concludes higher production of nitrogen containing OA and then we hypothesized this could be a possibility in our case. This was explained in more detail at the beginning of our response.

L487: Section 2.2.1 includes description of photooxidation without additional VOCs –

AUTHORS RESPONSE: We thank reviewer for pointing out the typo. This will be corrected to be 2.2.2

L505-507: the explanation doesn't seem to be valid. Why can't it be that the SOA form these VOCs has the same optical characteristic as the SOA formed in the absence of the VOCs?

AUTHORS RESPONSE: We agree that the wording here is strange. We will rephrase it in the revision as: "This is most likely because we took our measurements after 12

hours of aging, which was enough time for the scattering nature of the SOA produced during aging to drive measured SSA values to unity, regardless of the chemical pathway taken due to the addition of VOCs"

L539: it appears that there was insignificant additional SOA formed from oxidation of aromatics that were added to the chamber. What was the NOx level in the experiments? Perhaps the high NOx conditions of the burns lead to low SOA yield from these precursors and therefore no significant SOA is observed. If results from aging in the presence of additional VOC were not conclusive, I suggest removing all the discussion related to it throughout the paper. –

AUTHORS RESPONSE: NOx concentrations were only available during the chamber characterization experiments used for our previous paper. Equipment malfunctions prevented us from measuring this during chamber experiments. However, no additional NOx was added to the chamber during these experiments, only what was available due to combustion.

L 516-517: I'm not following why continuous size-selection was not possible during these aging experiments

AUTHORS RESPONSE: Unfortunately, our current setup does not allow continuous monitoring when particle sizes are being selected. Mainly, the volume of the nephelometer is too large compared with the relatively small flow rate provided by the WCPC. We have found that a flush time of at least $15 - 20$ minutes is required to completely replace a sample within the nephelometer with a new one (when changing particle sizes, for example). At this rate, it takes about 4 hours to collect measurements for three different mobility diameters, including blank measurements, repeat measurements, and the time to flush between samples.

* * *
**Fig. 1.** Temperature profile of the chamber. Runs 1b and 4b were done on the same day as runs 1a and 4a, respectively, with initial temperatures higher for 1b and 4b than 1a and 4a.

**Fig. 2.** Normalized particle size distribution for 500o C and 800o C burning cases

Legend:
- Dark
- UV
- UV + VOC
- 500 C Burn
- 800 C Burn

Y-axis: Geometric Mean Diameter (nm)
X-axis: Aged Time (hr)

**Fig. 3.** Same as Figure 1 in the manuscript (x axis in log scale).

---

## Author Comment (AC2) · 15 Apr 2020

A pdf copy of the response is also attached RESPONSES to REFEREE #2

This manuscript presents measurements of single scattering albedo (SSA) of size-selected aerosols emitted from controlled combustion of African biomass fuels under three conditions: fresh emissions, dark aged aerosols and photo-chemically aged aerosols. Three types of wood fuels were combusted in a tube furnace at two different temperatures (500 C and 800 C) and an indoor smog chamber was used to age

aerosols in clean and polluted (VOC rich) environments.

AUTHORS RESPONSE: We thank the reviewer for the time he/she took to provide us with valuable suggestions. The authors feel that the reviewer either misunderstood our descriptions and overlooked some aspects of the paper. We will show this is indeed true in our detailed responses with relevant references to each question and comment. We feel that there is rush to dismiss and devalue the work and undermine the paper instead of giving us a chance to respond and clarify our claims. We hope our responses and explanations will convince the referee to make a different final determination.

The authors claim that the significance of their work lies in providing optical and chemical characterization of a previous unstudied group of fuels that contribute significantly to aerosol emissions in Africa. However, there are no novel findings reported in this study and claims of significance are greatly overstated. While the particular fuels in this study might not have been characterized, there is a robust body of literature regarding the effect of combustion conditions on optical properties of emitted aerosols in controlled (example: Chen and Bond, 2010, ACP; Saleh et al., 2018, ES&T) as well as representative household use (Roden et al., 2006, ES&T; Chen et al., 2012, ES&T) settings. This study was limited by a lack of chemical characterization and SSA measurements limited to mid-visible wavelengths, and therefore could only reiterate the well-known effect of combustion temperature on absorption efficiency.

AUTHORS RESPONSE. There is no doubt that emissions from African continent is a major source of atmospheric aerosol, but their optical properties are least studied compared to emissions from other regions. Even though there are several studies on the impact of burning conditions on optical properties, we don't have full understanding as to why certain fuels burn with higher efficiency. For example, savannah grass fires tend to be of flaming type while boreal fires tend to be of smoldering type (Saleh et al., 2014). In addition, optical properties (such as absorption per mass) also show fuel dependency (Chen and Bond, 2010). All these facts suggest that we need more studies on different unstudied group of fuels. A recent critical review by Hodshire et

al. (2019) recommends the following: "(1) More smoldering fires within the laboratory setting should be studied. As well, methods to more closely match field burns should be developed, such as burning more compact fuels/larger amounts. . . (2) A wider sampling of fuels as laboratory studies have been focused predominantly on fuels that may exist in North American fires." This paper address both recommendations. Regarding all the examples provided by the reviewer, work by Chen and Bond, 2010 was done at lower temperatures (max 360 oC) whereas Saleh et al. 2018 was done for propane not for a biomass fuel. So, we think more studies are needed on biomass combustion. This work is presented as a two-part study, where this part focused on optical properties and chemical composition and characterization is presented in Part II, which is published in ACPD and we will refer the reviewer to the second part for a detailed discussion(Smith et al., 2020). Further integration of the two manuscripts will be done, and summaries from Part 2 will be incorporated into the revised manuscript.

Following the paragraph contrasting eucalyptus and acacia combusted at 800o C, the following text will be added to line 393:

"In the companion paper to this (Part 2), methanol extracts from BBA collected on Teflon filters were analyzed by ultra-performance liquid chromatography interfaced to both a diode array detector and an electrospray ionization high-resolution quadrupole time-of-flight mass spectrometer (UPLC/DAD-ESI-HR-QTOFMS) in negative ion mode. This was used to determine the relative abundance and light-absorption properties of biomass burning organic aerosol constituents. MS analysis of BBA extracts from combustion at 800 °C revealed very little difference between the two fuel types, suggesting that there are either very few BrC species produced for either fuel under these combustion conditions, or there are numerous species that are essentially the same between the samples. However, given that Eucalyptus has a higher SSA than Acacia, this would suggest that Eucalyptus has more non-absorbing OA, or at least less absorbing than BC. Since it is Acacia that appears to have many more low-abundant organic constituents, several possibilities exist to explain these differences in SSA, as explored in

more depth in Part 2. It is likely that Eucalyptus combustion products are not captured by some aspect of the extraction and UPLC/DAD-ESI-HR-QTOFMS analyses, that the observed differences in SSA are due to morphology differences, or some combination thereof. One potential explanation would be the presence of significant amounts of eucalyptol in the BBA, which is a large fraction of Eucalyptus oil, and is a cyclic ether that lacks any basic functionality amiable for negative ion mode analysis, has good solubility in alcohols, and does not absorb in the UV and visible. An examination of the UV-Visible spectra from the DAD shows no absorbing species in either region."

The following paragraph will be included in the manuscript following the discussion of fresh emissions produced by combusted at 500o C, added to line 410:

"Chemical analysis revealed that, when combusted at 500 °C, eucalyptus and acacia had a variety of compounds in common, such as lignin pyrolysis products, distillation products, and cellulose breakdown products. Several lignin pyrolysis products and distillation products are more prevalent in Eucalyptus than Acacia, while pyrolysis products of cellulose and at least one nitroaromatic species were more prevalent in Acacia. Given that these lignin pyrolysis and distillation products are known chromophores and are more prevalent in Eucalyptus than Acacia, while Acacia has a higher abundance of non-chromophores derived from sugars and cellulose, one would assume that Eucalyptus would be more absorbing in the visible (i.e. have a lower SSA) than Acacia. Despite the chemical analysis not capturing absolute amounts of OA, Acacia was found to have an SSA that is higher than Eucalyptus by 0.1 to 0.2, which is consistent with chemical measurements. This suggests that Acacia has either larger absolute amounts of non-chromophore compounds or Eucalyptus has a greater quantity of chromophores whose absorptive properties extend to the 500 – 570 nm region of the visible spectrum. An analysis of the chromatographically-integrated UV/Visible spectrum shows that there are chromophores whose absorption features peak ~290 nm and extend into the 500 – 570 nm region, though a normalized spectrum does not appear to show drastic differences between species."

Contrary to the reviewers comment we believe that our claim is not overstated for several reasons. (1) Most current optical properties measurements are limited to a single or a few selected wavelengths. The accurate measurement of aerosol optical properties over the entire solar spectrum is a technological challenge. Our system allows measurement of optical properties at a wide range of wavelengths over most of the solar spectrum to determine "featured" absorption cross sections as a function of wavelength. While most of the example provided by reviewer are limited to single wavelength our study was done for 500-570 nm range, and it is within our capability to conduct extinction measurements from 400-800 nm at any wavelength interval by simply changing mirrors. This represents a significant effort with the current dataset and lets its novelty be its justification. We haven't come across any experimental set up for aerosol optical properties measurement that can accomplish this. (2) We have more control of the combustion process. The tube furnace enables us to study different burning stages by controlling the temperature and speed of burning by adjusting the flow of air into the furnace. We have not come across aerosol optical measurement system that uses a tube furnace with so much control of the burning condition, as most studies simply burn an amount of fuel amenable for their measurement devices. The burn system closest to ours is the one reported by Chakrabarty's group (Sumlin et al., 2018b) where a heating coil is used to control temperature. The main difference is uniformity of the temperature throughout the furnace as opposed to a localized temperature in a coil and the control of the gas mixture flow during the burning. We have elected to present a range of combustion conditions to cover a range of combustion scenarios that could depend on different combustion circumstances (wildfires, land clearing, etc.). (3) Soot generation was done using a tube furnace, which was attached to an indoor chamber, and samples were directly fed into the optical properties measurement system to measure the optical properties as a function of aging. While smog chambers are not new, such an integrated system is not common.

We are surprised that the reviewer overlooked all the experimental features which alone are significant. Moreover, we disagree with the reviewer that our study does

**[ACPD](https://www.atmospheric-chemistry-and-physics.net/)**
not have novel findings. 1) There has never been a laboratory study of BB aerosols from biomass fuels in the region that we are aware of. 2) We showed that by just changing burning temperature, the SSA of emitted aerosol differ very significantly for the same BB fuel. This was different than Saleh et al. 2018, which was done for propane not biomass fuel. We showed that nighttime aging (dark aging) increase the SSA of aerosol, which we think this is one of the very few studies to report that.

In that sense, this work represents attempt towards filling the gaps in our understanding of biomass burning aerosol optical properties in this under-sampled and ignored part of the world. Africa is the single largest continental source of BB emissions, with recent studies estimating that it makes ∼55% of the global contributions to BB aerosols (Ichoku et al., 2008;Roberts et al., 2009;Roberts and Wooster, 2008;Lamarque et al., 2010;van der Werf et al., 2010;Schultz et al., 2008). Measurement of optical properties of biomass fuels from the region is long overdue.

The aging experiments show that both dark and photochemical aging reduce the absorption efficiency of size-selected aerosols (photochemical more so than dark) but no chemical properties were measured to illuminate the mechanism of absorption loss.

AUTHORS RESPONSE: In the accompanying paper, we analyzed methanol extracts from Teflon filters, as detailed in the excerpt above. Given that this is the main subject of Part 2, it wouldn't be appropriate to go into significant depth here, and we would refer the referee to the accompanying paper that is on ACPD for a detailed discussion. While measurements of dark aged samples were not performed, a summary of results from Part 2 regarding the light aged BBA will be provided and incorporated into the manuscript. With regards to photochemical aging, the text will be amended at line 504, as described two comments down from this.

Further, the aging results are only presented for 500 C aerosols because (Line 471): "Therefore, due to the very low number concentration and highly absorbing nature of the particles, the scattering coefficient at 800 C was below the detection limit of our

[Figure]

nephelometer during the aging experiments." The authors propose that future studies will include these missing measurements (performed by increasing the amount of fuel burnt) but I am puzzled why these changes were not made for this study.

AUTHORS RESPONSE: We tried the aging experiment with higher load and were not successful due to the limitation in the amount of fuel we can burn in our system at a single time. Furthermore, our current setup prohibits bulk aerosol measurement. This work is focused on conducting measurements for size selected particles. It is possible to perform these measurements for the entire size distribution, but this brings its own challenges and other parameters, such as changes in the size distribution itself, become unconstrained.

There are similar problems with aging experiments in a polluted environment (Line 505: "This is because we took our measurements after 12 hours of aging, which seems long enough to characterize the impact of the added VOC due to aging in UV. This fact suggests that a more carefully controlled study is needed to accurately simulate the impact of urban pollution on aerosol single scattering albedo") that indicate that the authors did not rigorously handle their motivating hypotheses, leaving glaring holes in their manuscript.

AUTHORS RESPONSE: This is hardly the case, though the authors admit that this could be phrased better. Starting at line 504 with the sentence "We attempted. . .", the remainder of this paragraph and the next (after Figure 7) will be replaced with the following text:

"Despite the use of anthropogenic VOCs, a concentration larger than those average values found in urban and suburban regions of South Africa, no distinct effect was observed for SSA values of BBA produced during combustion at 500 °C. While it is possible that the relatively long aging time could obscure some of the effects due to the presence of VOCs, it is also possible that combustion products dominate molecular species and the effects of additional VOCs are insignificant. The later would suggest

that anthropogenic pollution does not seem to affect the optical properties of BBA. Indeed, in examining the effects of aging on the chemical composition of BBA shows very few species that could be attributed to anthropogenic VOCs; specifically, only dihydroxyphthalic acid produced from xylene. For both Eucalyptus and Acacia, an isomer of dihydroxybenzene, such as resorcinol or catechol, was removed to the highest degree from the fresh BB aerosol upon photochemical aging. Generally, very few compounds were produced to a significant extent and both fuels were dominated by loss of chromatophoric lignin pyrolysis and distillation products. Not surprisingly, the associated absorbance from these chromophores, mostly from 200 – 350 nm, also attenuated with respect to age. This may be caused in part by the photo-bleaching effect created by irradiation of UV light for 12 hours, heterogeneous OH oxidation, and SOA formation of non-chromophores. This fact suggests that a study with a higher temporal resolution is needed to simulate the impact of the VOCs on aerosol SSA, where continuous or much more frequent measurement are needed to determine impact of urban pollution on aerosol single scattering albedo. Such a study, using continuous measurements, is not possible for our setup when particles are also size selected. Like dark aged conditions, we were not able to estimate SSA for combustion at 800 oC due to the low particle concentration and highly absorbing nature of the aerosol. A chemical analysis of aged BBA produced at this temperature revealed very few changes, suggesting there are few molecular species produced by combustion at this temperature."

Indeed, this is substantiated later in the paper by the lack of OA enhancement from the presence of anthropogenic VOCs.

Aside from concerns about significance and study design, there are significant issues in how the manuscript is presented. Instances of grammatical errors and confusing sentence construction are far too many to enumerate but more importantly, several arguments/claims are not supported by findings in this study or citations from literature.

AUTHORS RESPONSE: We thank reviewer for constructive comment and pointing out the grammatical errors. There is indeed no excuse for this and we will thoroughly

review the manuscript to fix the grammatical issues and clarify confusing sentences.

The authors establish that the fuels studied here are household fuels and acknowledge the potential differences between typical household use and controlled burning. They do not present any discussion of how findings from controlled combustion can be extended to a more realistic condition: this undermines the purported importance of their findings.

AUTHORS RESPONSE: We thank reviewer for valuable comment. ÂăWhile laboratory studies provide opportunities for control over the environmental and chemical conditions, so we can examine the effect of changing single variable at a time, a limited number of fuel-specific comparisons can be made between the laboratory and field studies given the lack of adequate field studies in Africa. Recreating the atmospheric conditions of plumes in the field, is still a challenge though this is a long-term goal of our research group. We acknowledge that there are significant knowledge gaps that limit a full understanding of how field and laboratory observations can be reconciled. This should not be a reason not to do laboratory experiments, but rather the reason to do more of those so we can clearly understand the physics and chemistry that governs the observed variability. However, we added following discussion on L 98: "This study was done in more controlled environment and to extend the result from this kind of study to more realistic condition we apply our optical property result to previously proposed parameterization schemes."

Further, they designate their 800 C burn condition as flaming (a reasonable assumption) and 500 C burn as smoldering (which is much higher than smoldering temperatures in literature). These assumptions are not substantiated with any further evidence. The SSA values reported for smoldering appear too low for pure smoldering combustion (eg. - those in (Sumlin et al., 2018a) and I am not convinced that the authors ensured that they are not from mixed combustion conditions.

AUTHORS RESPONSE: We agree with reviewer. There is no clear single line that

separates the smoldering and flaming fires, and it is likely that there is a gradient of conditions as a function of temperature. The designation was made based on the observed colors of the filter samples and measured MCE's. The latter is, of course, quantitative and enables the comparison of this work to others. Changes will be made throughout the manuscript with 800 oC described as "flaming-dominated burning" and 500 oC as "smoldering-dominated burning". As evident from the estimated MCE, flaming dominated and smoldering dominated are valid descriptions.

Line 415 (comparing SSA values here with previous studies) states: "This could explain why our SSA calculations for BrC was lower than expected". All measurements in this study are for total aerosols, BrC is mentioned without any justification.

AUTHOR RESPONSE: We appreciated reviewer comments. We will replace BrC with the phase "smoldering-dominated aerosol" throughout the manuscript. We agree that aerosol emissions at 500 oC have contributions from black carbon and other organic/inorganic components.

Many hypotheses are presented for the aging observations however the study was not conducted in a way that allows any plausible claims about "night-time formation or aromatic nitrogen containing compounds", for example. SOA formation is presented as a hypothesis for SSA reduction (Line 492) during photochemical aging but fragmentation of absorbing aerosols is not considered.

AUTHORS RESPONSE: We thank reviewer for constructive comments. The reason for making hypothesis for SOA formation during dark aging was that the SSA increase in those experiments was driven by an increase in scattering cross section with no evidence of changes in absorption cross section, as mentioned in the manuscript (L 455-458). There is lots of evidence of nighttime chemistry resulting in relatively high amounts of nitrogen containing SOA (Hartikainen et al., 2018;Tiitta et al., 2016;Li et al., 2015). That is the reason we made a statement about this possibility for our experiments. We acknowledge that this is a rich subject area, and there could be a variety of

chemical transformations, such as acid-catalyzed reactions. Regarding reviewer comments on photochemical aging, we also talked about the possibility of fragmentation of absorbing aerosol. As stated on Line 494-95 "An increase in SSA is possible during photochemical aging due to degradation in brown carbon absorptivity (Sumlin et al., 2017). So, we think we were not sticking only to an SOA hypothesis. If the reviewer is referring to the physical fragmentation of aerosol, the authors are not familiar with this process, nor do we consider it very likely. If the reviewer is referring to "collapse" of aggregates into shapes that are more spherical, it's very unlikely that such large changes in SSA can be attributed to that process.

Finally, the choice of figure type for representing the results in figures 4, 6 and 7 is baffling to me: why are SSA values plotted over this very narrow range of wavelengths? Clearly, no wavelength dependence can be seen between 500 and 570 nm. I fail to see the purpose of multiple figures that contain a series of zigzagging flat lines.

AUTHORS RESPONSE: Clearly, the purpose of those figures was to see the wavelength dependency, or lack thereof in this case. As previous studies (Sumlin et al., 2018a) showed that BrC has an impact on SSA with lower SSA at shorter wavelengths, though these measurements were performed at discrete wavelengths (like 405, 532, 600 etc.) and it was not quite clear that this range of wavelengths definitively shows a BrC effect on SSA. Most of the previous studies were limited to either 405, 532, 660, 780 nm measurements and the behaviors in between those wavelengths is typically assumed. Assuming something like a power-law dependence cannot realistically hold for a wide range of wavelengths. That's why we present whole range of SSA measurements between 500 to 570 nm which may appear as zigzag lines. Previous work in our laboratory shows that, even at this wavelength range, a wavelength-dependent SSA can be observed, and it should not be taken for granted or assumed that the range is too narrow for a wavelength-dependent conclusion to be made; only that the results are restricted to the most intense portion of the solar spectrum.

* * *

---

## Author Response (AR2)

**Reviewer #1**

I appreciate the efforts undertaken by the authors to revise the manuscript and respond to the reviewers' concerns. Although the paper has gone under major revisions, some of my concerns from before are still not fully addressed. I therefore cannot accept the manuscript and suggest major revisions. Personally, I believe it's a pity that the full extent of the chemical analysis results are not presented along with the optical properties of the aerosols since explanation of the optical properties is highly dependent on the chemical nature of the aerosols. I strongly encourage the authors to combine the two papers together. Please see below my comments:

**Author's response**:
 *We thank the reviewer for the earlier very constructive comments that helped improve the paper and the current comments will surely improve the paper further. With respect to combining the papers describing the optical and chemical properties, the authors initially considered that and felt the paper would be too long and too complicated, and decided to present it in two parts. The review process on part 2 is almost complete and the comments provided are positive. It is very likely that the second part will be published about the same time as part 1 if accepted.*
 *From our perspective as authors, we considered both papers as different aspects of a whole work. However, we did not fully appreciate the perspective of the reader, since it would be hardly fair to expect them to read both papers to understand one of them alone. As such, we thank the reviewer for their insight on this matter. As a result, we have added more results of the chemical analysis in this paper and provided several figures and tables in the supplementary documentation as discussed below in response to each comment. This additional material is not simply copied from Part 2, but tailored to the work in question.*
 *Note also the line numbers used by the reviewer refer to the line numbers on the marked up (track change) version. we will be using the line numbers of the clean copy.*

**General Reviewer Comments**:

- Discussion on the contribution of multiply charged particles is still confusing and not consistent; the authors first indicate that the error is not significant for 300 nm and 400 nm particles considered in this work; then they explain the procedure using an APM and conclude that the error in SSA for fresh 200 nm and 300 nm particles is ~8%. In the next paragraph, they indicate the error is more significant for the flaming conditions, but do not indicate if the 8% estimate includes flaming aerosols and if not, what the error for such conditions are. This discussion still needs to be cleared up and be consistent and complete for all sizes and conditions considered here.

**Authors Response:**
 *We thank the reviewer for the constructive comments. We rewrote both paragraphs (lines 355-372) and described in detail the issue of multiply charged particles and Mie theory calculation.*

**Changes made in the manuscript**

"Submicron aerosol show size-dependent SSA values in visible wavelengths. Size-selected SSA values measured at 532 nm in this study are compared with the size selected SSA values calculated using Mie theory for two different refractive indices, as shown in Figure 3. These refractive indices were proposed by Bond and Bergstrom (2006) for BC and Levin et al. (2010) for smoldering biomass burning particles. While no pronounced size dependence was observed for aerosol from flaming-dominated combustion, contrary to what was predicted by Mie theory, the SSA does show size dependence for aerosol from smoldering-dominated combustion. This is due to the impact of multiply charged aerosol not discriminated by the DMA. Since we did not correct for the presence of multiply charged particles in this work, as described in section 2.5, this behavior was expected. The impact of multiply charged particles is more significant for flaming-dominated combustion (no pronounced size-dependent SSA) compared with the smoldering-dominated combustion (with some size-dependent SSA). This is a consequence of difference in particle size distributions. As shown in **Figure S3**, the presence of a second peak in the size distribution for flaming-dominated burns was expected to increase the impact of multiply charged particles on the observed.

Recently, we estimated the impact of multiply charged particles on SSA using an Aerosol Particle Mass Analyzer (APM; Kanomax model 3602). We didn't have this capability early in this study. APM was connected in-line after the DMA and subsequent optical properties were measured for freshly emitted aerosol. Details about the measurement strategy is given by Radney and Zangmeister (2016). It was found that, for smoldering-dominated burns, our SSA values were over estimated by a maximum of 8% for 200 and 300 nm sizes and by 5% for 400 nm size particles. Whereas, for flaming-dominated burns, our SSA values were overestimated by 12% for 200 and 300 nm sizes and by 9% for 400 nm size.

**Reviewer comment**

- The increase in temperature during the first hour of aging is significant and will impact partitioning of the semivolatile components. This effect needs to be discussed.

**Author's response**

*We agree with the reviewer that the increase in temperature impacts partitioning of semivolatile compounds. The particle/gas partition coefficient, Kp is temperature dependent (Pankow, 1987) and its value is needed to predict the unknown gas-phase SVOC concentration from its measured concentration in airborne particles. However, Kp values can only be retrieved from the literature for a limited number of SVOCs. Furthermore, for SVOCs, the time to reach equilibrium partitioning between gas-phase and airborne particles varies between seconds to days depending on the volatility of the SVOCs and the diameter of the particle (Weschler and Nazaroff, 2008;Wei et al., 2016)). Ranjan et al. (2012) studied the gas–particle partitioning from POA emissions from diesel engines and showed that the results vary by about a factor of 4 across the atmospherically relevant range of temperature and organic aerosol concentrations confirming the semivolatile nature of POA emissions.*

*We did not measure the partitioning of the semivolatile components in this work. It is possible that this may impact the optical properties measurements following photochemical aging. The changes in optical properties could be attributed to be in part because of this.*

**Changes made in the manuscript**:

*We will include the discussion below on the partitioning of the semivolatile components due to temperature increase during photochemical aging qualitatively in section 2.2*

"The temperature increase during photochemical aging can impact partitioning of the semivolatile components. The gas phase partitioning coefficient is temperature dependent though its value is only available for a limited number of semivolatile compounds. We did not measure the impact of temperature on partitioning of the semivolatile components, but it is possible that the observed change in optical properties during photochemical aging could in part be attributed to this effect."

**Reviewer comment**

Regarding the Mie calculations…. It seems the authors just calculated scattering and absorption (or extinction) cross sections of the aerosols, assuming an overall RI based on RI of BC and bulk aerosol from Levin et al. In that case, what was the overall RI that includes contributions of BC and other components of the freshly emitted aerosols and how was this RI estimated? In other words, if RI of BC is combined with another RI, what weighting factor is given to BC vs. other aerosol components. The text should also clearly demonstrate that an inherent assumption here is that aerosols are internally mixed at all sizes such that the bulk estimate of RI is the same for all sizes. A discussion on how good such an assumption is (or is not) should also be provided.

**Author Response**

*We are sorry for the confusion about the SSA calculation using Mie theory. We will provide a more detailed explanation:*

- *In Figure 3, the symbols (square, diamond and circle) are the estimated SSA values from this study based on the size-dependent scattering and extinction measurements.*
- *We calculated the size dependent SSA for refractive index 1.95- i 0.79 using Mie theory (Black line in Figure 3).*
- *We calculated the size dependent SSA for refractive index 1.6 - i 0.1 using Mie theory (Red line in Figure 3).*

*Since we are not combining RI of BC and bulk aerosol, there is no need for making any assumption on mixing state, weighting factor, etc. We used Mie theory twice for the two different cases to calculate the SSA for two different values of the refractive index.*

**Changes made in the manuscript**

*As stated above the following sentence will be added in the manuscript to clarify this point*

"Submicron aerosol show size-dependent SSA values in the visible wavelength range. Size-selected SSA values measured at 532 nm in this study are compared with the size-selected SSA values calculated using Mie theory for two different refractive indices, as shown in Figure 3. These refractive indices were proposed by Bond and Bergstrom, (2006) for BC and Levin et al. (2010) for aerosol produced in smoldering biomass burning."

**Reviewer comment**

- Effect of NOx on SOA from aromatics: although NOx data are not available for these experiments, I think the authors should indicate that if NOx concentrations were high, it's possible

that yields from the injected aromatics were very small and that can explain why there wasn't any change in the photochemical aging results with and without VOCs. This is important to highlight before hypothesizing that anthropogenic pollution doesn't affect BB aerosol optical properties.

**Author Response:** *We thank the reviewer for this suggestion. This will be addressed, as follows in section 3.5.*

**Changes made in the manuscript**

"While it is possible that the relatively long aging time could obscure some of the effects due to the presence of VOCs, it is also possible that molecular species are dominated by combustion products, and the effects of additional VOCs are insignificant. The later would suggest that the three-aromatic species representing anthropogenic pollutants do not seem to affect the optical properties of BB aerosol. Another possibility would be that high NOx concentrations would prevent the formation of ozone, which would hinder SOA formation from aromatic species."

**Reviewer comment**

- There still seems to be a disconnect between the chemical analysis results and optical properties discussed here. Please see one of my comments below.
- L410(L420): When discussing the chemical analysis results, only that of acacia and eucalyptus are discussed. Wasn't analysis for the olive fuel carried out as well?

**Author Response:**

*Analysis for the olive fuel was not carried out. Unfortunately, time and funding did not allow us to carry out this analysis. Additionally, a three-way comparison would be significantly more complex.*

**Reviewer comment**

- L650 (L421): what's the basis of this statement: "Since it is Acacia that appears to have many more low-abundant organic constituents"? This goes back to my point that the composition data cannot be separated from this paper or else, the paper seems incomplete.
- Can authors include the uv-vis spectrum of the extract for different fuels and different burn conditions?

**Author Response**:

*The supporting information for this statement is now included in SI and is **Figure S5**. Identification for compounds with an absolute difference greater than 0.5% have been labeled and are listed in **Table S1**. References to this in the text have been made.*

**Reviewer comment**
- Can authors include the uv-vis spectrum of the extract for different fuels and different burn conditions?

**Author Response**:

*The authors have done this in **Figure S6**. References to this in the text have been made. Unlike in part 2, where the entire available wavelength range is discussed, the figures presented here focus on the 500-570 nm region of the spectrum.*

**Reviewer specific comments:**
- L719 (L461): change "at night" to "in the dark"

**Author Response:** *The change is made.*

**Reviewer comment**
-L-720 (L462) It will be valuable to show the change in aerosol size and density for the dark aging experiments in a plot.

**Author Response:**
      *Unfortunately, we did not possess the APM at the time these experiments were made, so density measurements are not possible. For the size distributions, this is provided as heat maps for number density as a function of time for smoldering-dominated Acacia and Eucalyptus as they aged in the dark. This is **Figure S1** and has been referenced in the text.*

**Reviewer comment**

L745 (L497): nitrites or nitrates?

**Author Response**: *nitrates. This will be specified in the text.*

**Reviewer comment**
L720-745 (L464-470): since RH in the current experiments was low and not changing, the discussion about chemistry under higher RH seems irrelevant.

**Author Response**
      *The discussion about chemistry under higher RH was included since the focus of the work is on African biomass fuels, in a region where RH is high near the tropics. While our chamber RH was, low and was not representative of the tropics, we plan to conduct measurements at high RH as a follow up of this project.*

**Reviewer comment**
- L793-794 (L498-499): This phrase is confusing: "(~12% by mass compared to ~12% from OH radical NO3 and ~76% from and proxy radical).

**Author Response**: *We have attempted to remedy this confusion.*

**Changes made in the manuscript**

"Only in the slow ignition case was there a significant formation of SOA from ozonolysis during dark aging (~12% by mass from ozonolysis compared to ~12% from OH radical $NO_3$ and ~76% from and proxy radical based on positive matrix factorization analysis)."

**Reviewer comment**

L920 (L543) I disagree; there are some chromophores that could be absorbing at longer wavelengths. Some of the chemical analysis discussion mentions presence of chromophores; if those analyses show only absorption at lower wavelengths, why include those results with the current study that focuses on extinction and scattering at 500-570 nm? How can the SSA of acacia and eucalyptus at these longer wavelengths be put in the context of the different types of chromophores that have been observed at lower wavelengths?

**Author Response**

*In this paragraph, we are discussing the current state of knowledge in the field, and in this sentence, we were merely stating that this work does not cover the same wavelength range as previous studies. We would expect absorption features to extend to this region of the spectrum from shorter wavelengths, as demonstrated in the spectra we present.*

**Changes made in the manuscript**

"These studies were made at wavelengths of 375 and 405 nm, while ours were done in the visible range, so they are not directly applicable to this work."

**Reviewer comment**

- L962 (L568): anthropogenic pollution is a very broad term. What was tested was only a mixture of 3 aromatic species. Please also see my comment above about NOx levels and SOA yields.

**Author Response**

*The reviewer is correct: indeed, anthropogenic pollution is too broad to use in this work as we are using few examples of pollutants.*

**Changes made in the manuscript**
*The text will read*
"The later would suggest that the three-aromatic species representing anthropogenic pollutants do not seem to affect the optical properties of BB aerosol."

**Reviewer comment**

- L 1008 (L601): What uncertainties about SMPS are they authors referring to? Can they be quantified?

**Author Response**

*We were referring to the uncertainty in the counts measured by the CPC in the SMPS system. According to TSI, the uncertainty in the particle count by the WCPC is ±10%.*

**Changes made in the manuscript**
*The following sentence is added*
"The uncertainty in the particle count by the WCPC is ±10%"

**Reviewer #2**

**Suggestions for revision or reasons for rejection (will be published if the paper is accepted for final publication)**

The study presents data on optical properties of biomass burning aerosols and on the effects dark- and photochemical aging on optical properties. The novelties of the study are the usage of sub-Saharan biomass fuels and the inclusion of the dark aging conditions in the experiments. The fact that dark aging affects optical properties in different way than photochemical aging is an important finding. Unfortunately, in the manuscript the experiments are quite poorly described both with respect to combustion conditions and chamber experiments. The combustion experiments are performed in a tube furnace using 0.5 g fuel pieces, which offers a good control over the combustion conditions. On the other hand, such a setup differs significantly from any real-world combustion appliance, both with respect to the combustion system geometry and the size of the fuel batch, which likely affects aerosol formation substantially. Therefore, a careful description the combustion conditions would be essential to make the results useful for the scientific community. Presently only MCE, which is a very simple metric depending on the exhaust CO concentration, is presented. The term "combustion temperature" is used throughout the manuscript without defining it (see detailed comments below). I would recommend to report at least the following data from the combustion experiments to make the data of this article more useful.
- Air-to-fuel ratio
- time-dependent CO, CO2 and hydrocarbon concentrations during combustion of the fuel batch
- Adiabatic combustion temperature

Moreover, the description of the experimental conditions in the chamber seem to be very limited. Clearly presented information on the concentrations of O3, non-methane VOC and NOx in the chamber (e.g. in a table) would be essential to understand the aging experiments. For UV-induced aging experiments, determination of the OH-radical exposure would be a benefit to relate the findings with other chamber experiments. It also seems that all experiments were not successful (e.g. concentrations were too low for some instruments to obtain data) and the authors conclude that further studies are needed.

The paper includes lot of discussion on the effects of particulate chemical composition on its optical properties. The discussed UPLC/DAD-ESI-HR-QTOFMS analyses of chemical constituents in the aerosols are very interesting and potentially very important for the scientific community. However, it is quite strange that no chemical data is presented in this paper and there are just citations to the "Part2-paper" where obviously, some of the discussed findings on chemical species are presented. In the current form, it is difficult for the reader to find the data since no figures or tables are referred to. In my opinion, if the chemical data is to be published in another paper, it would be more logical to publish first the chemical data. In any case, the discussed chemical data must be clearly visible for the readers. Overall, the paper addresses an important topic and presents potentially scientifically valuable results, but has some significant shortages in description of the experimental conditions and aerosol chemical compositions.

**Author Response:**
*The authors are grateful for this reviewer's comments and pointing out the need for some clarification on engineering aspects of combustion theory and thermochemistry related to this work. We believe the comments and our responses improve the quality of the paper for publication in ACP. We address the comments one by one*

- *The combustion apparatus is described in a later response.*

- *Hydrocarbon concentrations were not available during these experiments. However, in lieu of hydrocarbon measurements, we have made a plot for the temporal evolution of total particle mass in μg/m$^3$ assuming density of 1 gcm$^{-3}$ , CO, and CO$_2$ concentration (both in ppmv) for smoldering-dominated Acacia and Eucalyptus. This is **Figure S2** and is referenced within the text. These measurements are following the smog chamber, so any high temporal resolution phenomenon would be lost upon mixing within the chamber.*

- *Regarding NOx and ozone, we were unable to measure those species during these experiments due to instrumentation issues. Based on the work of other researchers, (Akagi et al., 2012) we would expect no initial ozone production. The measurements mentioned in the manuscript were measurements done during the chamber characterization experiments.*

- *Regarding the lack of presented chemical data, we agree that this should not have been omitted and have taken great lengths to amend this shortcoming. A comparison of mass spectra for fresh emissions from Acacia and Eucalyptus are presented in **Figure S5** and **Table S2**. The effects of aging are demonstrated for smoldering-dominated Eucalyptus (**Figure S9a and Table S3**) and Acacia (**Figure S9b and Table S4**).*

**Changes made in the manuscript**
*The following clarification is added in the manuscript*

"The measurements of O$_3$ and NOx were done only during the chamber characterization experiments and were not measured in subsequent experiments reported in this work**"**

**Reviewer comment**

More specific comments are below:

Line 102: The authors state that they "have full control of temperature, airflow, and material combusted." If that is the case, more details on the combustion process should be available, than the single value of MCE, which has been provided in this paper. Moreover, the term "combustion temperature" is used throughout the manuscript without defining it (Pro tip: the tube furnace set point temperature is not necessarily the same as the combustion temperature). In general, during batch combustion of biomass there are many parameters affecting the composition of biomass burning PM, including, but not limited to:
- air-to-fuel ratio
- combustion rate
- fuel moisture content
- fuel particle size

- combustion chamber materials and geometry (these will influence the e.g. radiative heating of the fuel sample from the combustion chamber walls)
- residence time of the flue gases in the combustion appliance
- exhaust gas dilution temperature and dilution ratio
It would be important to include such information for example in the supplementary material

**Authors Response**

*Our combustion apparatus has been thoroughly described in our previous characterization paper (Smith et al., 2019). Briefly, the furnace (Carbolite Gero, HST120300-120SN) holds an 85 mm OD, 80 mm ID, and 750 mm long quartz working tube and has a heated region of 300 mm. Stainless steel mounts and ceramic insulation plugs on either end enable the introduction and sampling of gasses. The furnace is preheated before the introduction of biomass samples. These samples are placed in a quartz combustion boat (AdValue Technology, FQ-BT-03), weighed and pushed into the center of the furnace with the aid of tongs, before replacing the input insulator and flange. The smoke and gasses produced from combustion are sent directly to the chamber via a heated (200 °C), and ¼-inch OD stainless steel bellows transfer tube. The combustion boat was 100 mm long, 41 mm wide, and 20 mm tall. **The authors do not feel this level of detail is required in the current work, as the information is already readily available in a published form**.*

*We calculated that the residence time of combustion gases within the tube furnace to be ~9.2 seconds. The dilution temperature and dilution ratio is not applicable, since a diluter was not used. Effluent from the tube furnace was sent directly to the smog chamber."*

*The authors agree that the term "combustion temperature" may be misleading. In this work, we mean the set point temperature of the furnace when the sample is first placed into the furnace. **This will be reworded in Section 3.1 of the manuscript.** Further, we believe there is some confusion regarding the purpose of these experiments. We are using a furnace (described above) to ignite the fuel for generating biomass burning aerosol. We do not attempt to measure the properties or capability of the biomass as a fuel for use in cook stoves or combustion engines. While we agree with the reviewer that several of these quantities could be gathered from this data, we are unaware of any other research groups in atmospheric science using these values for biomass burning aerosol research. The paragraphs below are an attempt to provide this data for the reviewer, but we have chosen not to include this in the final manuscript since it is beyond the scope of our work.*

*Although we do not have the exact elemental composition for the species used in this work, we can approximate them using data from Table 1 of Silva et al. (2019). For our purposes, the "branch" category most closely resembled what we burned in our lab. Acacia has a C:H:N:O:S ratio (by composition) of 0.483 : 0.0582 : 0.0228 : 0.4352 : 0.00067, while Eucalyptus has a respective ratio of 0.4412 : 0.0525 : 0.0133 : 0.4926 : 0.00035. This gives a stoichiometric air-to-fuel ratio of 3.18 for Acacia, and 2.28 for Eucalyptus. This assumes ideal conditions where the fuel is completely combusted. The actual air to fuel ratio also requires complete measurement of hydrocarbons, total particle mass, and elemental particle composition which we did nott have.*

**Changes made in the Supplementary Document**
*The following information will be provided in the supplementary document.*

The combustion time for our fuels was between 5 and 10 minutes. The exact time is unknown since we could not see into the furnace during combustion. The ash left on boat was less than 3% of the fuel mass.

Heat transfer to wood particles is directly proportional to the exposed surface area of the sample. Smaller particles have a larger surface area to mass ratio than larger particles. This in turn means a faster rate of heat transfer to the particles and, consequently, a faster combustion rate. (Simmons, 1983). In this work, small twigs were used for combustion, fuel particle size can be approximated as a cylinder of mass 0.5 g, with an average radius of 0.25 cm and an average length of less than 5 cm, giving an upper limit of 0.0012 $m^3$ for the volume, an upper limit of 0.000825 $m^2$ for the exposed surface area of the fuel, and a lower limit of 0.405 $kg/m^3$ for the density of the fuel used. Typically, whole twigs with a thin layer of bark were used, although occasionally two or more smaller pieces may have been used to achieve the same mass. Occasionally, these twigs were also split open. These factors could increase the surface area of the fuel by approximately 0.000325 $m^2$ for the same fuel mass.

We calculate that the residence time of combustion gases within the tube furnace is ~9.2 seconds. The dilution temperature and dilution ratio is not applicable, since a diluter was not used. Effluent from the tube furnace was sent directly to the smog chamber.

**Changes made in the manuscript**:

*We will clarify the definition of "combustion temperature"* on Lines 341-343. It should now read: "Each fuel was placed into the tube furnace set at two different temperatures (initially heated at 500 ℃ and 800 ℃) to investigate the impact of ignition temperature on aerosol optical properties and chemical composition. In both cases, the furnace was allowed to reach the desired temperature before the sample was placed inside."

"The moisture content of the biomass fuel samples was 10%."

**Reviewer comment**

Line 119: "0.5 g of fuel was burned normally, which produced about 600 to 800 μg m-3 of mass loading in the chamber". As the chamber volume is 9 m3, the particulate mass derived from the 0.5 g batch is up to 7200 ug, which equals to particulate emission factor of 14400 mg/kg fuel. This is a quite high emission factor for any combustion system. The authors should discuss the relevance of this result in the paper.

**Authors Response**

*We thank the reviewer for the comments. We don't think 14400 mg/kg (14.4 g/kg) of emission factor is quite high for smoldering fires. For example, Liu et al. (2017) reported average EFs of PM1 as 26 ± 6.2 g/kg in western US wildfires. In addition, Akagi et al. (2011) reported PM EFs for Boreal, Temperate, and Extratropical forests as larger than 15 g/kg. So, we are confident that the ranges of EFs we are getting for smoldering dominated burns are within the ranges estimated by past studies as reported in the current literature.*

**Reviewer comment**

Line 135: "For this work, only zero air was used at a flow rate of 10 sL min–1."
What was the air-to-fuel ratio? Why air was used, although there was a possibility to use lower oxygen contents?

**Authors Response:**
*As mentioned in a previous response, the air-to-fuel ratio was between 2.28 and 3.18 for the fuels used in this study. Reducing the oxygen content is certainly within our experimental capacity. However, it has not yet been explored as a variable. We found that temperature alone was enough to distinguish between smoldering and flaming combustion for our experiments.*

**Reviewer comment**

Line 139: "the tube furnace provides a uniform temperature throughout the sample."
What is the basis for this argumentation? How much is the heat released by the burning fuel particle in comparison to the heat fluxes from the tube furnace walls? The adiabatic combustion temperatures could be calculated based on the fuel composition, air-to-fuel ratios and external heat from the tube furnace walls.

**Authors Response:**
*This statement was based on the tube furnace specification, as there is a temperature gradient in the tube that provides a uniform heated region of 300 mm, in which the sample boat resides. The concept of Adiabatic combustion or flame temperature is generally used as an engineering design criterion for air –firing combustion chambers and power plants. This quantity is not generally reported in aerosol research and we fail to see the relevance to the work reported here.*

*This quality is generally easy to calculate for gaseous fuels. Information on the composition of liquid or solid fuels is generally much more limited than that for gaseous fuels. Physical and chemical properties of a wood, such as chemical composition and thermal properties, play an important role in its combustion. However, based on available information we have calculated the adiabatic flame (combustion) temperature and was 3706 °C and 3679 °C for acacia and eucalyptus, respectively.*

.

**The following is included in the supplementary information**:
The adiabatic flame (combustion) temperature is 3706 °C and 3679 °C for acacia and eucalyptus, respectively.
**Changes made in the manuscript:** *The following sentence is added to clarify the temperature uniformity.*
"as the tube provides a uniform heated region of 300 mm which is approximately the size of the quartz boat."

**Reviewer comment**

Line 193: "Aerosol growth in the chamber was expected to be due to coagulation, diffusional losses of particles," This is already mentioned on the line 184. -> the factors related to particle

growth are unnecessary repeated.

**Author's Response.** *The reviewer is correct. The text is edited to remove repetition.*

**Reviewer comment**
Line 206: Please considering putting the section "2.1.3 Chamber cleaning" into the supporting material, as it consists only of technical details on the implementation of the experiments.

**Author's Response:**
     *We agree with the reviewer. Chamber cleaning and other additional experimental details are included in the supplemental section.*

**Reviewer comment**

Line 222: "Optical properties were measured using the procedure described below soon after the chamber was well mixed." This is not a very informative sentence.

**Authors Response:** *We agree with the reviewer.*

**Changes made in the manuscript**: *The text will be rewritten to read:*
"Optical properties for fresh samples were measured using the procedure described below within 90 minutes of combustion, which allowed the size distribution of the aerosol to stabilize enough to conduct measurements without having a significant change in number density over the course of the experiment."

**Reviewer comment**

Line 224: "For the photochemical aging, a new burn was made, and the particles were kept in the chamber for 12 hours with the UV lights on." What does it mean that a new burn was made? How is this experiment different from the dark aging, apart from the fact that the UV-lights were on, remains unclear.

**Authors Response**
     *For the dark aged experiments, the sample was left in the chamber after fresh measurements were made. For photochemically aged experiments, a new sample was needed, since the original sample had already been altered by dark aging. Due to the length of time necessary to make fresh measurements, these were not conducted for the photochemically aged sample; instead, the UV lights were turned on immediately after combustion, and the sample was allowed to age for 12 hours before taking any measurements. The reviewer is correct that the only difference in the aging experiments is the presence of UV radiation, which is the only variable that changes. The manuscript will be edited for clarity.*

**Changes made in the manuscript**: *The text will be rewritten to read:*
"For photochemical aging, a fresh sample was introduced into a clean chamber with the UV lights on immediately after combustion. Measurements were made after 12 hours of UV radiation.*"*

**Reviewer comment**

Line 229-: I suggest to start the paragraph by explaining how the experiment "in a polluted environment" is different to an experiment "clean environment". In the present form the reader patiently has to go through lot of technical details related to measurement accuracy etc. before even knowing how these experiments were performed. It also remains unclear whether the VOC:s were injected into the chamber before or after feeding the BB-smoke sample? How did adding of VOC:s affect the NMVOC:NOx -ratio in the chamber? Are the NMVOC:NOx -ratios representative to the air of the South African sites in question?

**Authors Response:**
*We agree with the reviewer about the introduction to this section. As mentioned in the response to a previous comment, hydrocarbon concentrations were unavailable for these experiments.*

**Changes made in the manuscript**: *The following paragraph will be added at Line 229:*
"To study the effect of photochemical aging in a polluted environment, a mixture of VOCs was used to simulate an urban atmosphere. These VOCs were injected into the chamber and allowed to mix with the chamber air while the furnace heated up before the introduction of fuel samples. The rest of this section describes the preparation of this VOC mixture."

**Reviewer comment**

Line 397-399: It is not possible to define the chemical composition of particles, based on the MCE. MCE is a very simple metric depending on the exhaust CO concentration. Generally, there is no good correlation between CO and OC/BC in biomass combustion. In my opinion the authors should present measured data on BC and OC concentrations to state such findings. The color of the filter is not a very scientific method for chemical characterization of aerosol particles.

**Authors Response**
*We agree with the reviewer that correlation of CO is not very good with OC/BC. But for biomass burning emissions, for lower MCE (< 0.9), the fraction of BC is very low (below 10% or so), whereas for higher MCE ( >0.98), the fraction of BC is significantly higher. So, we still think we can say that less BC is produced during smoldering combustion, whereas more BC is produced during flaming combustion as in previous studies (Lewis et al., 2008;Liu et al., 2014;Pokhrel et al., 2016;McMeeking et al., 2014;Stockwell et al., 2016) We agree with the reviewer that color of the filter is not a scientific method for chemical characterizations but we can clearly say that if the color of the filter is black then the emission is dominated by the BC; otherwise it will not look black. The intent for showing the filters was illustrative, not quantitative.*

**Changes made in the manuscript**: *The text will be rewritten to read:*

"A qualitative visual measure of the impacts of combustion temperature on aerosol properties can also be gleaned by looking at the color of the collected filter samples, as shown in Figure S4."

**Reviewer comment**

Line 430-433: "Chemical analysis revealed that for smoldering-dominated combustion, Eucalyptus and Acacia had a variety of compounds in common, such as lignin pyrolysis products, distillation products, and cellulose breakdown products. Several lignin pyrolysis products and distillation products are more prevalent in Eucalyptus than in Acacia, while pyrolysis products of cellulose and at least one nitroaromatic species were more prevalent in Acacia." Please refer to a table or figure where this information can be seen in a numerical form.

**Authors Response:**

*This information is now provided in supplementary information. Figures (difference mass spectra) are provided in Figures S5 and S9. The accompanying tables (S2, S3, and S4) give the quantitative information for each peak, and suggested identities. Additionally, for each suggested identity, there is a compound type listed (lipid, lignin pyrolysis product, distillation products, sugar/cellulose product, nitro-aromatic compound, oxidized polyaromatic hydrocarbon, and oxidized anthropogenic volatile organic compound).*

**Reviewer comment**

Line 467: "BB aerosol was aged without UV lights on and kept overnight for 24 hours."
This is not much information about the conditions during the experiment. What was the ozone concentration during the dark aging period? Was O3 added into the chamber? Later (Line 481) it is argued that oxidation of organic aerosol was initiated by O3. Is there no measured information on O3 ? That would be a notable shortage to describe the experimental conditions.

**Authors Response**

*We thank the reviewer for constructive comments. As expected, ozone is a secondary product formed in biomass burning smoke(Akagi et al., 2012) and there was no ozone initially. Furthermore, during dark aging, production of ozone is not expected in the absence of radiation.*

**Changes made in the manuscript**; *Line 467 now read as "*BB aerosol was aged in dark for 24 hours in absence of UV lights and additional ozone*". We edited Line 481 which now reads as "*The potential mechanism for the observed result could be due to the formation of less/non-absorbing secondary organic aerosol".*

**Reviewer comment**

Line 570-573: Data on the concentrations of chemical species in the chamber should be presented in this paper to discuss the findings.

**Author's Response**:

*Regarding the chemical constituents discussed in this section (i.e. those in the particulate phase), this information has been provided, as indicated earlier. References to SI have been included in the text.*

**Reviewer comment**
Line 598-599: "Our connecting tubing was short enough (0.5 m) to neglect such a loss." What is the basis for this conclusion? The length of the tube cannot be used as a proof of low losses.

**Author's Response:**

*The tubing that transports the smoke between the burn chamber and the smog chambers is also susceptible to losses/ delays of gas-phase SOA-precursor material (Pagonis et al., 2017). Losses of ultrafine (those below 100 nm) particles inside the sampling tubes were observed, with smaller particles suffering greater losses (Kumar et al., 2008) Furthermore, they showed relatively greater losses of particles with increased length of sampling tubes. The basis of the conclusion is the citations mentioned Kumar et. al will be included in the revision.*

.

Completion Date: 6/30
*there may need to be a separate section on "changes to the manuscript" after the author response

[revised manuscript text omitted]